# LOTFormer: Doubly-Stochastic Linear Attention via Low-Rank Optimal Transport

## Abstract

Transformers have proven highly effective across a wide range of modalities. However, the quadratic complexity of the standard softmax attention mechanism poses a fundamental barrier to scaling them to long context windows. A large body of work addresses this with linear attention, which reformulates attention as a kernel function and approximates it with finite feature maps to achieve linear-time computation. Orthogonal to computational scaling, most attention mechanisms—both quadratic and linear—produce row-normalized maps that can over-focus on a few tokens, degrading robustness and information flow. Enforcing doubly-stochastic attention alleviates this by balancing token participation across rows and columns, but existing doubly-stochastic attention mechanisms typically introduce substantial overhead, undermining scalability. We propose LOTFormer, a principled attention mechanism that is simultaneously linear-time and doubly-stochastic. Our approach exploits the connection between attention maps and transportation plans between query and key measures. The central idea is to constrain the transport plan to be low-rank by conditioning it on a learnable pivot measure with small support. Concretely, we solve two entropic optimal transport problems (queries→pivot and pivot→keys) and compose them into a conditional (glued) coupling. This yields an attention matrix that is provably doubly-stochastic, has rank at most $r \ll n$, and applies to values in $O(nr)$ time without forming the full $n \times n$ map. The pivot locations and masses are learned end-to-end. Empirically, LOTFormer achieves state-of-the-art results on the Long Range Arena benchmark, surpassing prior linear and transport-based attention methods in both accuracy and efficiency.

## 1 Introduction

Transformers (Vaswani et al., 2017) have emerged as one of the most influential neural network architectures, achieving state-of-the-art performance across a wide range of domains (Lin et al., 2022), from natural language processing (Grattafiori et al., 2024) to computer vision (Khan et al., 2022), audio and speech processing (Gulati et al., 2020), multi-modality (Liu et al., 2023), protein folding (Jumper et al., 2021), and bioinformatics (Dalla-Torre et al., 2025). At the core of Transformer architectures lies the *attention* mechanism, which effectively models complex dependencies among tokens in a sequence. As the demand for models capable of reasoning over long contexts continues to grow, extending Transformers to larger context windows has become increasingly important. However, this goal is hindered by the quadratic computational complexity of the attention mechanism. This challenge has spurred a growing body of research aimed at reducing the computational burden of attention, with a prominent line of work focusing on *linear attention* methods (Katharopoulos et al., 2020; Wang et al., 2020; Choromanski et al., 2020; Shen et al., 2021; Chen et al., 2021; Xiong et al., 2021; Meng et al., 2025). Similarly, our primary goal is to design attention mechanisms with linear complexity in sequence length.

On the other hand, prior work has shown that the row-stochastic nature of attention matrices often leads to attention concentrating disproportionately on a few tokens, which can hinder effective information flow (Sander et al., 2022; Shahbazi et al., 2025). In the case of vision Transformers, this phenomenon—referred to as *token overfocusing*—has been observed to degrade performance, and promoting broader token participation has been found to improve both robustness and accuracy (Guo et al., 2023). Moreover, recent work demonstrates that mitigating overfocusing yields

smoother, more interpretable attention maps and stronger performance on dense prediction and object discovery tasks (Darcet et al., 2024). One approach to mitigate overfocusing is to transform the row-stochastic attention matrix into a doubly-stochastic one, for example, using the Sinkhorn algorithm (Sinkhorn, 1964). While effective in enhancing robustness, this approach incurs even greater computational cost than standard quadratic attention, since Sinkhorn iterations are applied on top of the full quadratic attention map. More recently, Shahbazi et al. (2025) exploited the connection between transport plans (Peyré & Cuturi, 2019) and doubly-stochastic attention matrices and, building on advances in efficient transport plan computation (Liu et al., 2025), introduced ESPFORMER—a more computationally efficient formulation of doubly-stochastic attention. Nevertheless, ESPFORMER remains quadratic during training due to its reliance on soft-sorting and exhibits super-linear complexity at inference.

In this work, we take a step further by investigating, for the first time, the feasibility of computing doubly-stochastic attention in *linear* time. We propose a novel attention mechanism that, similar to ESPFORMER, formulates attention as a transportation plan between the empirical measures defined by queries and keys. Our key innovation is the introduction of a learnable *pivot measure* with a small support size $r \ll n$, which serves as an intermediate representation. Instead of directly computing the $n \times n$ transport plan, we factorize it by first solving for the optimal transport between queries and the pivot, and then between the pivot and keys. This construction, which we denote as LOTFormer, yields a low-rank decomposition of the attention matrix that simultaneously preserves its doubly-stochastic structure and reduces the computational complexity from quadratic to linear in the sequence length. Figure 1 illustrates our framework for sample queries and keys under varying pivot sizes, i.e., different values of $r$.

**Our specific contributions** are summarized below:

1. We introduce LOTFormer, a mathematically rigorous mechanism for computing doubly-stochastic attention with linear complexity in sequence length.

2. We demonstrate that LOTFormer achieves state-of-the-art performance on the Long Range Arena (LRA) benchmark.

3. We show that LOTFormer achieves state-of-the-art performance on ImageNet-1K compared to other linear attention methods.

## 2 RELATED WORK

### 2.1 LINEAR ATTENTION MECHANISMS

The original Transformer architecture relies on the quadratic-time softmax attention, which quickly becomes prohibitive for long sequences (Vaswani et al., 2017). To mitigate this issue, a large body of work has introduced *linear attention* mechanisms that approximate or restructure the attention computation. Early efforts include *Local Attention* (Aguilera-Martos et al., 2024) and *Longformer* (Beltagy et al., 2020), which restrict interactions to a local context window to reduce complexity. More structured sparsity patterns are adopted by *Reformer* (Kitaev et al., 2020) and *BigBird* (Zaheer et al., 2020), which employ locality-sensitive hashing and block-sparse attention, respectively.

A second family of approaches replaces the softmax kernel with low-rank or kernelized approximations. *Linear Transformer* (Katharopoulos et al., 2020) and *Performer* (Choromanski et al., 2020) leverage kernel feature maps to re-express attention as a dot-product in a lower-dimensional space, reducing complexity to linear in sequence length. *Linformer* (Wang et al., 2020) introduces low-rank projections of the key–value matrices, while *Nyströmformer* (Xiong et al., 2021) applies Nyström approximations for kernel matrices. Similarly, *Kernelized Attention* (Luo et al., 2021), *Cosformer* (Qin et al., 2022), and *Skyformer* (Chen et al., 2021) extend this principle with improved kernel feature maps, orthogonal projections, or structured bases. *Synthesizer* Tay et al. (2021) departs from query–key interactions altogether by learning synthetic attention weights. Finally, *Informer* (Zhou et al., 2021) and *Hedgehog* (Zhang et al., 2024) address time-series and structured sequence modeling by combining efficient approximations with task-specific inductive biases. Together, these works establish the central paradigm of linear attention: replacing dense quadratic interactions with kernel, low-rank, or sparse mechanisms that preserve expressivity while improving scalability.

### 2.2 DOUBLY-STOCHASTIC ATTENTION

While linear attention addresses efficiency, another line of research has focused on the *stochastic structure* of attention matrices. The softmax operation enforces row-stochasticity, but not column

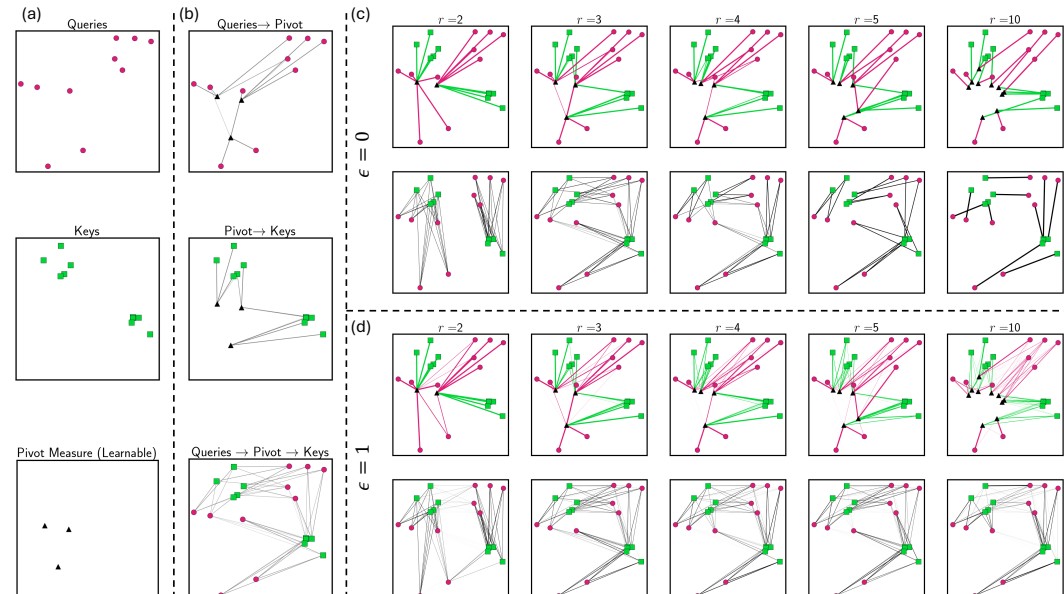

Figure 1: Illustration of LOTFormer. **(a)** Queries (red circles), keys (green squares), and the learnable pivot measure (black triangles). **(b)** Factorization of the transport plan: instead of solving directly for a full $n \times n$ coupling between queries and keys, LOTFormer first computes transport from queries→pivot and pivot→keys, then composes them into a glued coupling. **(c–d)** Effective query–key couplings induced by the pivot measure for different pivot sizes ($r = 2, 3, 4, 5, 10$). Top row of each block shows the mediated connections via pivots, and bottom row shows the resulting query–key couplings. **(c)** Without entropic regularization ($\varepsilon = 0$), couplings are sharp and sparse. **(d)** With entropic regularization ($\varepsilon = 1$), couplings become smoother and more diffuse.

normalization, leading to asymmetric attention maps. *SinkFormer* (Sander et al., 2021) addressed this by enforcing *doubly-stochastic* (DS) constraints via iterative Sinkhorn normalization, thereby producing balanced transport-like couplings between queries and keys. Building on this, *ESPFormer* (Shahbazi et al., 2025) leveraged *expected sliced transport plans* with annealed temperature schedules, achieving both theoretical guarantees and empirical improvements over SinkFormer by bridging soft and hard stochastic maps.

More recently, *Quantum DS Attention* (Born et al., 2025) has extended this framework into quantum-inspired settings, exploiting properties of quantum stochastic matrices and unitary constraints to model richer forms of normalization while preserving tractability. These approaches highlight that double stochasticity is not merely a regularization tool but a principled rethinking of attention as a transport problem, ensuring symmetry, stability, and interpretability in the learned couplings.

### 2.3 POSITION OF LOTFORMER

Our work, *LOTFormer*, sits at the intersection of these two threads. On the one hand, it inherits the efficiency benefits of linear attention by reformulating the attention kernel in terms of linearizable operations. On the other hand, it extends the doubly stochastic paradigm by embedding transport-inspired constraints into a linearized setting, thereby enabling scalable *doubly stochastic attention* with provable structure and practical efficiency. By unifying these perspectives, LOTFormer opens a new direction for scalable and structured attention that maintains both computational tractability and principled normalization.

## 3 METHOD

**Setup.** Let $\{x_i\}_{i=1}^n \subset \mathbb{R}^{d_{\mathrm{in}}}$ be the set of $n$ input tokens to the attention block and

$$q_i = W_Q x_i \in \mathbb{R}^{d_k}, \qquad k_i = W_K x_i \in \mathbb{R}^{d_k}, \qquad v_i = W_V x_i \in \mathbb{R}^{d_v},$$

for $i \in [1, n]$, denote the queries, keys, and values accordingly, with learned projections $W_Q, W_K \in \mathbb{R}^{d_k \times d_{\mathrm{in}}}$ and $W_V \in \mathbb{R}^{d_v \times d_{\mathrm{in}}}$. Let $Q = [q_1, \dots, q_n]^\top \in \mathbb{R}^{n \times d_k}$, $K = [k_1, \dots, k_n]^\top \in \mathbb{R}^{n \times d_k}$, and $V = [v_1, \dots, v_n]^\top \in \mathbb{R}^{n \times d_v}$.

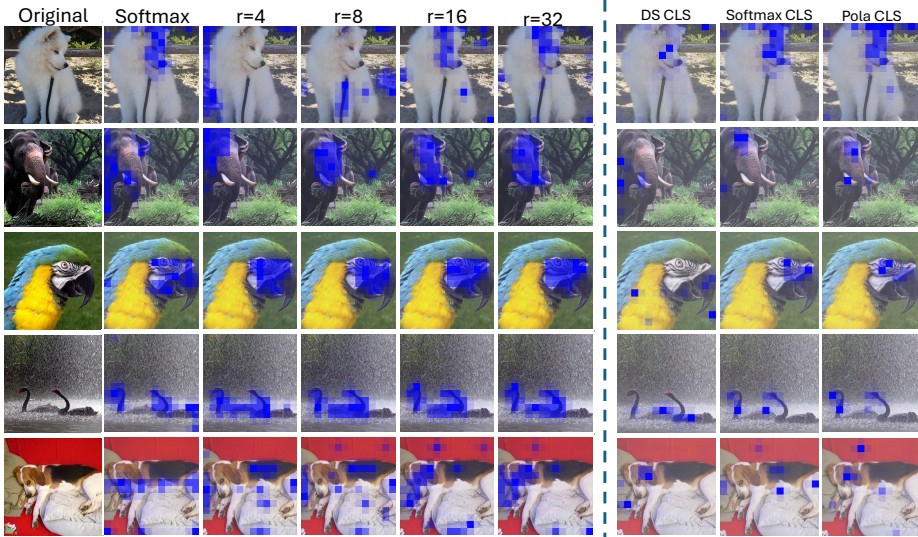

Figure 2: Patch-level visualizations of [CLS] attention. **(Left)** Comparison of standard SOFTMAX attention with LOTFormer at different pivot sizes $r \in \{4, 8, 16, 32\}$, showing how larger $r$ produces sharper, more object-centric maps. **(Right)** Effect of different [CLS] treatments (all without DWC): enforcing DS on [CLS] (*full DS*) degrades global aggregation, whereas decoupling it via [CLS]–softmax restores broad coverage, and adding polarization (+Pola) further sharpens selectivity. The leftmost column preserves the original image for context, while the other columns use a neutral gray background to standardize contrast.

**Empirical measures and pivot.** In line with earlier approaches that establish a connection between attention mechanisms and optimal transport (OT) (Sander et al., 2022; Shahbazi et al., 2025), we represent queries and keys as empirical measures, namely,

$$\mu = \sum_{i=1}^{n} \mathrm{p}_i^1 \delta_{q_i}, \qquad \nu = \sum_{j=1}^{n} \mathrm{p}_j^2 \delta_{k_j}, \qquad (\text{typically } \mathrm{p}_i^1 = \mathrm{p}_j^2 = \tfrac{1}{n}),$$

and introduce a *learnable pivot measure* with support on $r \ll n$ points:

$$\sigma = \sum_{t=1}^{r} \mathrm{p}_t^0 \delta_{z_t}, \qquad Z = \{z_t\}_{t=1}^{r} \subset \mathbb{R}^{d_k}, \ \ \mathrm{p}_t^0 > 0, \ \sum_{t=1}^{r} \mathrm{p}_t^0 = 1, \ \ r \ll n.$$

**Conditional OT** $\mu \to \sigma \to \nu$**.** An alternative but fundamental perspective on attention is that the entropically regularized transport plan between $\mu$ and $\nu$—that is, their joint measure—can be interpreted as a doubly stochastic attention matrix linking queries and keys. Our approach builds on this view by introducing a conditional transport plan, defined relative to a pivot measure, to construct such a coupling. By design, this formulation yields an attention matrix that is both low-rank and linearly decomposable.

Here we define the entropic OT problems from $\mu$ to $\sigma$, and from $\sigma$ to $\nu$. Let us use the dot-product similarity $c(q, k) = q^\top k$ and define

$$C_{it}^{(1)} = q_i^\top z_t, \qquad C_{jt}^{(2)} = k_j^\top z_t$$

With a shared regularization $\varepsilon > 0$, solve the two entropy-regularized transport problems

$$\Gamma^{(1)} \in \arg \max_{\Gamma \in U(\mathrm{p}^0, \mathrm{p}^1)} \langle \Gamma, C^{(1)} \rangle + \varepsilon H(\Gamma), \tag{1}$$

$$\Gamma^{(2)} \in \arg \max_{\Gamma \in U(\mathrm{p}^0, \mathrm{p}^2)} \langle \Gamma, C^{(2)} \rangle + \varepsilon H(\Gamma), \tag{2}$$

where $U(\alpha, \beta) = \{\Gamma \geq 0 : \Gamma \mathbf{1} = \alpha, \ \Gamma^\top \mathbf{1} = \beta\}$ and $H(\Gamma) = -\sum_{a,b} \Gamma_{ab}(\log \Gamma_{ab} - 1)$.

By classical OT theory, the maximizers of the above problems exist. Note that these transportation plans admit Sinkhorn scaling forms of:

$$\Gamma^{(1)} = \mathrm{Diag}(u) \, K^{(1)} \, \mathrm{Diag}(v), \quad K^{(1)} = \exp\big(QZ^\top / \varepsilon\big),$$

$$\Gamma^{(2)} = \mathrm{Diag}(\tilde{u})\, K^{(2)}\, \mathrm{Diag}(\tilde{v}), \quad K^{(2)} = \exp\!\big(KZ^\top/\varepsilon\big).$$

As $\varepsilon \to 0$, the entropically regularized transport plans $\Gamma^{(1)}$ and $\Gamma^{(2)}$ converge to their corresponding optimal transport plans. Having obtained $\Gamma^{(1)}$, the plan from $\mu$ to $\sigma$, and $\Gamma^{(2)}$, the plan from $\sigma$ to $\nu$, we now construct the conditional plan, also referred to as the "glued coupling," between $\mu$ and $\nu$.

**Glued coupling and doubly-stochastic attention.**    Define the glued (composed) coupling

$$\Gamma \;=\; (\Gamma^{(1)})^\top\, \mathrm{Diag}(\sigma)^{-1}\, \Gamma^{(2)} \;\in\; \mathbb{R}^{n\times n}. \tag{3}$$

Then $\Gamma\mathbf{1} = \mu$ and $\Gamma^\top\mathbf{1} = \nu$, i.e., $\Gamma$ is a transportation plan between $\mu$ and $\nu$. In the common balanced case $\mu = \nu = \frac{1}{n}\mathbf{1}$, we set

$$A \;=\; \Gamma \;=\; (\Gamma^{(1)})^\top\, \mathrm{Diag}(\sigma)^{-1}\, \Gamma^{(2)}, \qquad A\mathbf{1} = \mathbf{1}, \;\; \mathbf{1}^\top A = \mathbf{1}^\top,$$

so $A$ is *doubly stochastic*. The LOTFormer head output is

$$\mathrm{LOTAttn}(Q,K,V) \;=\; A\,V \;=\; (\Gamma^{(1)})^\top\Big(\mathrm{Diag}(\sigma)^{-1}\,(\Gamma^{(2)}V)\Big). \tag{4}$$

**Rank and complexity.**    Write

$$A \;=\; \big((\Gamma^{(1)})^\top\, \mathrm{Diag}(\sigma)^{-1/2}\big)\,\big(\mathrm{Diag}(\sigma)^{-1/2}\Gamma^{(2)}\big),$$

hence $\mathrm{rank}(A) \leq r$ (Note, in default setting $r \leq n$, we can show $\mathrm{rank}(A) = r$). We may avoid forming $A$ itself by using its factorized form, first computing $Y = \Gamma^{(2)}V \in \mathbb{R}^{r\times d_v}$, then $Z' = \mathrm{Diag}(\sigma)^{-1}Y$, and finally $O = \Gamma^{(1)}Z' \in \mathbb{R}^{n\times d_v}$. Each Sinkhorn iteration uses matrix–vector scalings with kernels $K^{(1)} = \exp(QZ^\top/\varepsilon)$ and $K^{(2)} = \exp(ZK^\top/\varepsilon)$, costing $O(nr)$ per iteration (plus $O(nd_k r)$ to form $QZ^\top$ and $ZK^\top$ once per update). For fixed $r \ll n$ and modest iteration count $T$, the per-head complexity is $O\big(nd_k r + T\,nr\big)$, i.e., *linear in $n$*.

**Learning.**    The pivot locations $Z = \{z_t\}$, masses $\sigma \in \Delta^{r-1}$, and projections $(W_Q, W_K, W_V)$ are trained end-to-end by backpropagating through the shared-$\varepsilon$ Sinkhorn problems equation 1–equation 2 and the glued composition. Note that for multi-head attention, each head has its own $(Z, \sigma)$.

**Why not the optimal Low-Rank OT?**    Our construction produces an attention matrix that is, by design, low-rank due to the introduction of the pivot measure. However, this should not be conflated with the *optimal low-rank transport plan* studied in prior work Scetbon & Cuturi (2022); Scetbon et al. (2021). In Low-Rank OT, the support of the low-rank factors is optimized directly to minimize the transportation cost, yielding a low-rank approximation that is provably optimal in that restricted class. By contrast, in our setting, the pivot measure is learned end-to-end as part of the attention mechanism, but not optimized explicitly for transport cost minimization. This distinction naturally raises the question of why not simply employ Low-Rank OT instead? The key reason is computational: computing Low-Rank OT still requires access to the full $n \times n$ cost matrix between $\mu$ and $\nu$ as input, which entails quadratic time and memory. This completely undermines the goal of linear-time attention. In contrast, our conditional transport construction never forms the full cost matrix, but instead only evaluates $QZ^\top$ and $ZK^\top$, each of size $n \times r$ with $r \ll n$, achieving linear complexity in $n$ while preserving double-stochasticity.

## 4 EXPERIMENTS

### 4.1 RUNTIME AND WALL-CLOCK ANALYSIS

We study the computational efficiency of LOTFORMER, a doubly–stochastic *linear-time* attention, against representative quadratic and linear baselines. The quadratic group includes *Softmax*, *ESP-Former (SoftSort)*, *ESPFormer (HardSort)*, and *Sinkformer*. The linear group includes *Performer*, *Nyströmformer*, *MobiAttention*, and *PolaFormer*. To expose the speed/accuracy control that low-rank methods afford, we also plot LOTFORMER at ranks $r \in \{16, 32, 64, 128, 256\}$; $r{=}64$ is shown with a solid line and the other ranks with dashed lines.

Figure 3 reports forward-pass runtime (ms/iteration) versus sequence length on a log–log scale for $N \in \{2^{9:17}\}$. The left panel compares quadratic methods to LOTFORMER; the right panel compares

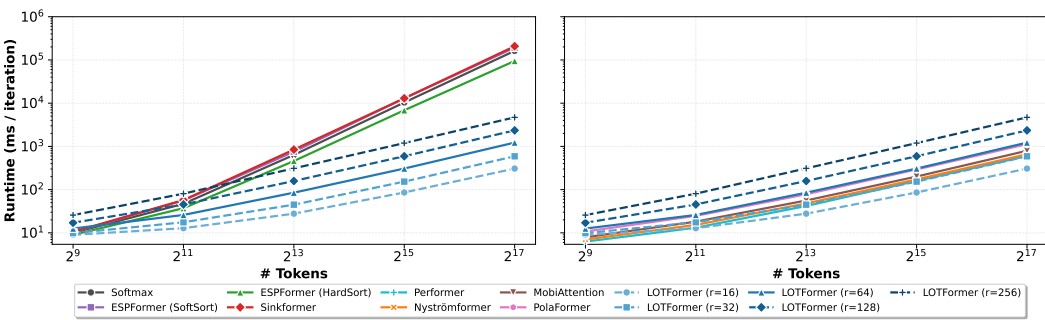

Figure 3: **Runtime scaling with sequence length.** Forward-pass runtime (ms/iteration) vs. $N$ on a log–log scale for quadratic methods (left) and linear methods (right). LOTFORMER is shown at ranks $r \in \{16, 32, 64, 128, 256\}$ in both panels (solid for $r=64$, dashed otherwise). Points denote measured values at $N \in \{2^{9:17}\}$;

linear-time baselines to the same LOTFORMER curves. As expected, Softmax and the DS variants exhibit near–quadratic growth in $N$, whereas LOTFORMER and the linear baselines grow roughly linearly. Across LOTFORMER ranks, lower $r$ yields faster curves and higher $r$ yields slower curves, illustrating a clear rank–time trade-off.

## 4.2 LONG RANGE ARENA

We evaluate our approach on the *Long Range Arena (LRA)* benchmark, a widely used suite for testing the ability of sequence models to capture long-range dependencies across diverse modalities, including structured parsing (ListOps), natural language understanding (Text), image classification (Image), document retrieval (Retrieval), and synthetic reasoning (Pathfinder). Following PolaFormer (Meng et al., 2025) and FlattenAttention (Han et al., 2023), we optionally apply a channel-wise 1D depthwise convolution over the token dimension to the value stream prior to mixing:

$$\tilde{V} = \text{DWConv}_k(V), \qquad O = A\,\tilde{V},$$

where $A$ is the attention map (DS for non-`[CLS]` rows; `[CLS]` via its head). DWC injects a local inductive bias and improves token mixing at negligible parameter/FLOP cost.

**Results.** We evaluate LOTFormer on LRA with $r = 32$ both with and without DWC and present our results in Table 1. It can be seen that, without using a DWC, our method achieves the second highest average performance, beaten only by Polaformer Meng et al. (2025) which utilizes a DWC. By adding the DWC, we significantly boost our performance on both the text and image tasks, achieving the best average performance among all methods while also achieving the best performance on the image task and second highest performance on the text task.

## 4.3 IMAGENET-1K

Next, we evaluate LOTFormer on ImageNet-1K with a DeiT-Tiny (Touvron et al., 2021) backbone and compare against recent linear-attention variants (Table 2). When applying doubly stochastic attention in ViTs, special care is needed for the `[CLS]` token, since it must act as a hub that aggregates information from all other tokens; a naive application may restrict its ability to gather global context. In addition, many recent linear-attention variants achieve improvements through orthogonal techniques that are compatible with LOTFormer. For example, PolaFormer and FlattenAttention employ lightweight depthwise convolutions (DWC) on values, while PolaFormer also introduces polarization on the logits— both of which could be seamlessly incorporated into our framework. In this section, we first discuss our special treatment of the `[CLS]` token, and then provide a thorough study on ImageNet-1K to disentangle the effects of these complementary techniques and our proposed LOTFormer. In short, we demonstrate that LOTFormer with a `[CLS]`-specific polarization head (Pol-`[CLS]`) and a lightweight DWC on values reaches 74.8% at negligible overhead (see ablations in Table 3), surpassing strong baselines such as PolaFormer (74.6%) at the same resolution.

| Model | Text | ListOps | Retrieval | Pathfinder | Image | Avg. |
|---|---|---|---|---|---|---|
| Transformer | 61.55 | **38.71** | 80.93 | 70.39 | 39.14 | 58.14 |
| LocalAttn | 52.98 | 15.82 | 53.39 | 66.63 | 41.46 | 46.06 |
| LinearTrans. | 65.90 | 16.13 | 53.09 | 75.30 | 42.34 | 50.55 |
| Reformer | 56.10 | 37.27 | 53.40 | 68.50 | 38.07 | 50.67 |
| Performer | 65.40 | 18.01 | 53.82 | **77.05** | 42.77 | 51.41 |
| Synthesizer | 61.68 | 36.99 | 54.67 | 69.45 | 41.61 | 52.88 |
| Longformer | 62.85 | 35.63 | 56.89 | 69.71 | 42.22 | 53.46 |
| Informer | 62.13 | 37.05 | 79.35 | 56.44 | 37.86 | 54.57 |
| Bigbird | 64.02 | 36.05 | 59.29 | 74.87 | 40.83 | 55.01 |
| Linformer | 57.29 | 36.44 | 77.85 | 65.39 | 38.43 | 55.08 |
| Kernelized | 60.02 | 38.46 | **82.11** | 69.86 | 32.63 | 56.62 |
| Cosformer | 63.54 | 37.20 | 80.28 | 70.00 | 35.84 | 57.37 |
| Nystrom | 62.36 | 37.95 | 80.89 | 69.34 | 38.94 | 57.90 |
| Skyformer | 64.70 | 38.69 | 82.06 | 70.73 | 40.77 | 59.39 |
| Hedgehog | 64.60 | 37.15 | 82.24 | 74.16 | 40.15 | 59.66 |
| PolaFormer$_{\alpha=3}$ | **73.06** | 37.35 | 80.50 | 70.53 | 42.15 | 60.72 |
| LOTFormer | $65.2 \pm 0.2$ | $38.5 \pm 1.2$ | $80.4 \pm 0.5$ | $73.2 \pm 0.9$ | $45.7 \pm 1.2$ | 60.6 |
| + DWC | $71.1 \pm 1.0$ | $38.5 \pm 1.4$ | $80.9 \pm 0.2$ | $69.9 \pm 0.5$ | **$54.1 \pm 1.5$** | **62.9** |

Table 1: Results on the LRA benchmark. The LOTFormer results are averaged over three runs. Best numbers per column are in bold. Our method achieves the highest overall average accuracy, with particularly strong gains on the Image task.

**[CLS] under DS attention.** In the extreme case (no entropy regularization and $r = n$), a doubly stochastic attention matrix can degenerate into a permutation, causing each query to attend to only one key—so the [CLS] token would aggregate values from just a single token. Although entropy regularization and low-rank constraints mitigate this, the issue persists in principle and conflicts with the [CLS] token's role as a global aggregator. To address this, we retain standard softmax aggregation for the [CLS] token, while enforcing doubly stochasticity only among image tokens. This preserves the classic softmax attention for [CLS]:

$$\alpha_{\mathrm{cls},j} = \frac{\exp\big(\beta \langle q_{\mathrm{cls}}, k_j \rangle\big)}{\sum_l \exp\big(\beta \langle q_{\mathrm{cls}}, k_l \rangle\big)}, \qquad \beta > 0,$$

which decouples [CLS] from the DS row–column coupling and restores true global aggregation, while all non-[CLS] rows remain DS.

**Polarization variants: Pol-[CLS] vs. Pol-All.** Following Meng et al. (2025), let $x^+ = \mathrm{ReLU}(x)$ and $x^- = \mathrm{ReLU}(-x)$. Pol-[CLS] (ours for LOTFormer) sharpens only the [CLS] interactions by forming polarized logits

$$\tilde{s}_{\mathrm{cls},j} = \big((q_{\mathrm{cls}}^+)^\top k_j^+ + (q_{\mathrm{cls}}^-)^\top k_j^-\big)^{p_s} + \big((q_{\mathrm{cls}}^+)^\top k_j^- + (q_{\mathrm{cls}}^-)^\top k_j^+\big)^{p_o},$$

followed by *softmax* over $j$: $\alpha_{\mathrm{cls},j} = \mathrm{softmax}_j(\tilde{s}_{\mathrm{cls},j})$. By contrast, Pol-All (as in PolaFormer) applies the same polarized-*logit* construction to *every* row $i$, then normalizes each row with *softmax*: $\alpha_{i,j} = \mathrm{softmax}_j(\tilde{s}_{i,j})$. Non-[CLS] rows in LOTFormer remain DS; only the [CLS] row uses softmax (or Pol-[CLS]+softmax). We reflect this choice in tables as Head/Pol $\in$ {Softmax, Pol-[CLS] (softmax), Pol-All (softmax)}.

**Depthwise convolution on values (DWC).** We study DWC in both PolaFormer (Pol-All) and LOTFormer (Pol-[CLS]) and observe consistent gains; our best LOTFormer uses Pol-[CLS] (softmax) + DWC.

**Reporting protocol.** Table 2 reports the best configuration per method at the same input resolution and discloses two implementation choices: the [CLS] Head/Pol (Softmax / Pol-[CLS] (softmax) / Pol-All (softmax)) and whether DWC is used. DS methods (Sinkformer, ESPFormer, LOTFormer) adopt [CLS]–softmax to enable global aggregation; only LOTFormer evaluates Pol-[CLS]. PolaFormer uses Pol-All and DWC. Detailed ablations isolate the effects of (i) the [CLS] head and (ii) DWC.

Table 2: DeiT-Tiny on ImageNet-1K with linear-attention variants. Best in **bold**; other top-3 are underlined. DS methods use [CLS]–softmax for the [CLS] row; Pol-* entries use *softmax* kernels (polarized logits + softmax).

| Method | Reso | Params | FLOPs | [CLS] Head/Pol | DWC | Top-1 (%) |
|---|---|---|---|---|---|---|
| DeiT (Touvron et al., 2021) | $224^2$ | 5.7M | 1.1G | Softmax | – | 72.2 |
| EfficientAttn (Shen et al., 2021) | $224^2$ | 5.7M | 1.1G | Softmax | – | 70.2 |
| HydraAttn (Bolya et al., 2022) | $224^2$ | 5.7M | 1.1G | Softmax | – | 68.3 |
| EnhancedAttn (Cai et al., 2022) | $224^2$ | 5.8M | 1.1G | Softmax | – | 72.9 |
| FLattenAttn (Han et al., 2023a) | $224^2$ | 6.1M | 1.1G | Softmax | ✓ | 74.1 |
| AngularAttn (You et al., 2023) | $224^2$ | 5.7M | 1.1G | Softmax | – | 70.8 |
| MobiAttn (Yao et al., 2024) | $224^2$ | 5.7M | 1.2G | Softmax | – | 73.3 |
| PolaFormer | $224^2$ | 6.1M | 1.2G | Pol-All (softmax) | ✓ | 74.6 |
| Sinkformer | $224^2$ | 5.7M | 1.2G | Softmax | – | 70.2 |
| ESPFormer ($L{=}8, \tau{=}0.1$) | $224^2$ | 5.7M | 1.2G | Softmax | – | 70.1 |
| LOTFormer ($r{=}32$) | $224^2$ | 6.1M | 1.2G | Pol-[CLS] (softmax) | ✓ | **74.8** |

Table 3: Side-by-side ablation on DeiT-Tiny / ImageNet-1K. "Pola type" indicates where polarization is applied: None, All tokens (PolaFormer), or [CLS]-only (LOTFormer). $\Delta$ is computed within each block relative to the previous row.

| PolaFormer | | | | | FlattenAttention | | | | | LOTFormer ($r{=}32$) | | | | |
|---|---|---|---|---|---|---|---|---|---|---|---|---|---|---|
| [CLS]-Softmax | Pola type | DWC | Top-1 (%) | $\Delta$ | [CLS]-Softmax | Pola type | DWC | Top-1 (%) | $\Delta$ | [CLS]-Softmax | Pola type | DWC | Top-1 (%) | $\Delta$ |
| ✓ | All | ✗ | 61.9 | 0.0 | ✓ | None | ✗ | 71.8 | 0.0 | ✗ | None | ✗ | 68.2 | 0.0 |
| ✓ | All | ✓ | 74.6 | +12.7 | ✓ | None | ✓ | 74.1 | +2.3 | ✓ | None | ✗ | 73.2 | +5.0 |
| | | | | | | | | | | ✓ | [CLS] | ✗ | 73.6 | +0.4 |
| | | | | | | | | | | ✓ | [CLS] | ✓ | **74.8** | +1.2 |

To clearly differentiate the effects of DWC and polarization, Table 3 contrasts PolaFormer, FlattenAttention, and LOTFormer under identical training settings. For PolaFormer, adding DWC to the value stream yields a large gain (+12.7 pts; 61.9→74.6), indicating that a lightweight local inductive bias complements Pol-All effectively. FlattenAttention also benefits from DWC (+2.3 pts; 71.8→74.1). For LOTFormer, gains are step-wise within the DS regime: enabling a [CLS] softmax aggregator provides the dominant lift (+5.0 pts; 68.2→73.2), switching to [CLS]-only polarization adds a smaller but consistent boost (+0.4 pts; 73.2→73.6), and inserting DWC yields a further +1.2 pts (73.6→74.8). Overall, DS attention benefits most from a dedicated [CLS] aggregator, with Pol-[CLS] and DWC providing complementary improvements at negligible cost.

**Effect of $r$.** Accuracy improves with larger pivot size $r$ and saturates around $r{=}32$–$64$. The Pol-[CLS] + DWC variant consistently outperforms the base DS configuration at each $r$, indicating that a specialized [CLS] aggregator and lightweight local mixing are complementary to the low-rank transport factorization. We use $r{=}32$ by default for DeiT-Tiny as it offers a strong accuracy–efficiency trade-off.

**Qualitative comparison of [CLS] attention.** In Fig. 2, the *left panel* shows the source image (first column), the standard *softmax* [CLS] map (second), and *LOTFormer* [CLS] maps for increasing pivot size $r$ (remaining columns, left→right). For small $r$, LOTFormer exhibits a *clustering* effect in which attention splits across several coarse regions; as $r$ increases, the maps become progressively *more object-centric*, concentrating on semantically relevant parts while retaining sufficient global coverage. The *right panel* contrasts [CLS] treatments—full DS, [CLS]–softmax (tokens-only DS), and +Pola, highlighting how each modulates selectivity and spatial support.

## 4.4 NEURAL MACHINE TRANSLATION

We evaluate LOTFormer against two recent attention mechanisms that also rely on doubly stochastic matrices, namely Sinkformer and ESPFormer. For fairness, we embed all three variants into two standard reference backbones: the Transformer and its DiffTransformer coun-

Table 4: Effect of pivot size $r$ for the best LOTFormer configuration (Pol-[CLS] + DWC) on ImageNet-1K (DeiT-Tiny). Forward/backward times are per training step (4×GPU setup).

| | $r{=}4$ | $r{=}8$ | $r{=}16$ | $r{=}32$ | $r{=}64$ |
|---|---|---|---|---|---|
| Top-1 (%) | 64.5 | 66.4 | 68.8 | **74.8** | 73.9 |
| Forward time (ms/iter) | 64.8 | 74.3 | 90.2 | 122.1 | 183.4 |
| Backward time (ms/iter) | 160.5 | 165.3 | 183.4 | 227.8 | 244.6 |

terpart Ye et al. (2025). Both backbones are implemented using the `fairseq` sequence modeling toolkit Ott et al. (2019) and trained on the IWSLT'14 German-to-English dataset Cettolo et al. (2014). Each backbone consists of a 6-layer encoder and a 6-layer decoder.

We adopt a two-stage evaluation protocol. In the first stage, we pre-train the baseline Transformer and DiffTransformer models for 25 epochs using the standard training procedure. We then conduct a *Plug-and-Play evaluation*, where the attention heads of LOTFormer, ESPFormer, and Sinkformer are directly inserted into the pre-trained models and evaluated without further training. The results of this evaluation are summarized in Table 5.

In the second stage, we perform a *Fine-Tune Boost* phase, where the Plug-and-Play models are further fine-tuned for an additional 10 epochs. Fine-tuning consistently improves performance across all three doubly stochastic variants. Notably, LOTFormer achieves the largest gains, surpassing both ESPFormer and Sinkformer on the Transformer and DiffTransformer backbones, and attaining the best BLEU scores of 34.64 and 34.83, respectively. These results demonstrate the effectiveness of LOTFormer as a transport-based doubly stochastic attention mechanism.

Table 5: Plug-and-Play and Fine-Tune performance on IWSLT'14 De→En (median over 4 runs). Note that $^\star$ indicates plug-and-play performance when swapping in a different attention than the base model's own. Moreover, $\Delta$ shows the change vs. the base model in each block.

| Model | Base: Transformer | | | Base: DiffTransformer | | |
|---|---|---|---|---|---|---|
| | Plug-and-Play | Fine-Tune | $\Delta$ | Plug-and-Play | Fine-Tune | $\Delta$ |
| Transformer | **33.40** | 34.61 | — | **33.85** | 34.78 | — |
| Sinkformer | 33.36$^\star$ | 34.61 | +0.00 | 33.67$^\star$ | 34.81 | +0.03 |
| ESPFormer | 33.38$^\star$ | 34.64 | +0.03 | 33.72$^\star$ | 34.83 | +0.05 |
| LOTFormer (ours) | 33.29 | **34.72** | **+0.11** | 33.42 | **34.91** | **+0.13** |

### 4.5 ABLATION STUDY

Here we ablate different design choices in our experiments and present them in Table 6. All ablation studies were carried out on ImageNet-100, and with a down-sized DeiT Tiny with halved hidden dimensions. We first compare a *fixed* vs. *learnable* ref with uniform mass. At tighter entropies ($\varepsilon=0.1$), performance improves steadily as $T$ increases, e.g., fixed: $79.6 \to 80.9$ (from $T=1$ to 20), learnable: $79.8 \to 81.1$, reflecting that *tighter $\varepsilon$ requires more Sinkhorn iterations to converge*. At a looser setting

Table 6: Ablation results for LOTFormer. Top-1 Accuracy (%) on ImageNet-100.

| **(i) Vary $T$ @ $\varepsilon \in \{0.1, 1\}$** | | | | | |
|---|---|---|---|---|---|
| | | $T=1$ | $T=5$ | $T=10$ | $T=20$ |
| fixed ref | $\varepsilon=0.1$ | 79.6 | 80.4 | 80.8 | 80.9 |
| | $\varepsilon=1$ | 80.2 | 81.2 | 81.4 | 81.7 |
| learnable ref | $\varepsilon=0.1$ | 79.8 | 80.7 | 81.1 | 81.1 |
| | $\varepsilon=1$ | 80.5 | 81.5 | **82.1** | 82.0 |

| **(ii) Vary $\varepsilon$ @ $T=10$** | | | | | |
|---|---|---|---|---|---|
| | 0.01 | 0.1 | 1 | 10 | 100 |
| fixed ref | 79.1 | 80.8 | 81.8 | 81.2 | 79.5 |
| learnable ref | 79.2 | 81.1 | **82.1** | 81.3 | 79.6 |

($\varepsilon=1$), both variants climb more quickly and *saturate earlier* (fixed: $80.2 \to 81.8$ at $T=10$, learnable: $80.5 \to$ **82.1** at $T=10$), with the learnable ref consistently ahead by $\approx 0.3$–$0.4$ points. The bottom panel (varying $\varepsilon$ at $T=10$) shows the same pattern from the complementary view: accuracy *peaks near $\varepsilon=1$* (fixed 81.8, learnable **82.1**) and drops at extremes (e.g., $\varepsilon=0.01$ or 100), indicating under-converged or over-smoothed transport when $T$ is held fixed. Overall, a *learnable* ref matches or exceeds a fixed one across settings and reaches the best value of **82.1**% at ($\varepsilon=1$, $T=10$).

## 5 CONCLUSION

We introduced LOTFORMER, a transport-based attention mechanism that is both linear-time and doubly stochastic. By conditioning on a learnable low-rank pivot measure and composing two entropic OT plans (queries→pivot and pivot→keys), LOTFormer yields a rank-$r$ attention map that applies to values in $O(nr)$ time without forming the full $n \times n$ matrix. Empirically, LOTFormer attains state-of-the-art results on the Long Range Arena and competitive ImageNet-1K performance, and further benefits from a `[CLS]`-specific polarization head and lightweight depthwise convolutions. We envision that this work will encourage the broader use of transport structures in scalable attention and motivate future directions on adaptive pivots, multi-scale pivots, and efficient training-time/inference-time schedules for Sinkhorn iterations.

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

## A NOTATIONS AND CONVENTIONS

**Vectors, matrices, and functions.**

- $\mathbb{R}^d$: $d$-dimensional Euclidean space.
- Vectors are column vectors unless stated otherwise.
- Matrices are uppercase Roman letters (e.g., $A \in \mathbb{R}^{m \times n}$).
- $\mathbf{1}_m$: all-ones vector in $\mathbb{R}^m$.
- $\mathrm{Diag}(a)$: diagonal matrix with entries of vector $a$.
- $|A| = \sum_{ij} A_{ij}$: sum of entries of matrix $A$.
- $\langle f, \mu \rangle$ where $f$ is a function, $\mu$ is a measure. The integration of $f$ with respect to $\mu$. That is

$$\langle f, \mu \rangle = \int f d\mu.$$

**Tokens and attention.**

- $X = [x_1, \ldots, x_n] \in \mathbb{R}^{n \times d_{\mathrm{in}}}$: input tokens (rows).
- $W_Q \in \mathbb{R}^{d_k \times d_k}, W_K \in \mathbb{R}^{d_k \times d_k}, W_V \in \mathbb{R}^{d_k \times d_v}$, weight matrices for Query, Key, Value
- $Q = XW_Q^\top \in \mathbb{R}^{n \times d_k}$, $K = XW_K^\top \in \mathbb{R}^{n \times d_k}$, $V = XW_V^\top \in \mathbb{R}^{n \times d_v}$: queries, keys, values.
- $q_i, k_i, v_i$: row vectors of $Q, K, V$.
- $\mathrm{LOTAttn}(Q, K, V) = AV$: attention output, with $A$ the (low-rank OT) transport matrix.

**Probability measures.**

- $\Delta_m = \{p \in \mathbb{R}^m : p_i \geq 0, \mathbf{1}_m^\top p = 1\}$: probability simplex.
- $\Delta_m^+ = \{p \in \Delta_m : p_i > 0\}$: strictly positive simplex.
- Empirical distributions: $\mu = \sum_i p_i^1 \delta_{q_i}$, $\nu = \sum_j p_j^2 \delta_{k_j}$.
- Pivot distribution: $\sigma = \sum_{t=1}^r p_t^0 \delta_{z_t}$ with $Z = [z_1, \ldots, z_r]^\top \in \mathbb{R}^{r \times d_k}$.

**Optimal transport.**

- $\alpha, \beta \in \Delta_m$: auxiliary probability masses for $U$ and $K$.
- $U(\alpha, \beta) = \{\Gamma \in \mathbb{R}_+^{m \times n} : \Gamma \mathbf{1}_n = \alpha, \Gamma^\top \mathbf{1}_m = \beta\}$: set of couplings.
- $H(\Gamma) = -\sum_{ij} \Gamma_{ij}(\log \Gamma_{ij} - 1)$: entropy.
- Entropic OT:
$$\Gamma^* \in \arg \max_{\Gamma \in U(\alpha, \beta)} \langle \Gamma, C \rangle + \varepsilon H(\Gamma).$$
- $\sigma = \sum_{i=1}^k \mathrm{p}_i^0 \delta_{z_i}$: auxiliary reference measure.
- $\Gamma^1 \in \mathbb{R}^{k \times n}$: optimal solution for $EOT_\epsilon(\sigma, \mu)$
- Glued coupling: $\Gamma = \Gamma^{(1)} \mathrm{Diag}(p^0)^{-1} (\Gamma^{(2)})^\top$.
- Balanced case: $p^1 = p^2 = \frac{1}{n} \mathbf{1}_n$, then $A = \Gamma$ is doubly stochastic.

## B RELATION BETWEEN LOW-RANK OT AND LINEAR OT.

**Notation setup and low rank optimal transport.** Given a matrix $\gamma \in \mathbb{R}_+^{n \times m}$, its nonnegative rank is defined by

$$rk_+(M) := \min\{q | M = \sum_{i=1}^q R_i, s.t. \forall i, \mathrm{rank}(R_i) = 1, R_i \geq 0\}$$

where $R_i \geq 0$ means each entry of $R_i$ is non-negative, $\text{rank}(R_i) = 1$ means that $R_i = a_i b_i^\top$ for some $a_i \in \mathbb{R}^n, b_i \in \mathbb{R}^m$.

In the discrete measure setting, i.e., $\mu = \sum_{i=1}^n p_i^1 \delta_{x_i}, \nu = \sum_{j=1}^m p_j^2 \delta_{y_j}$, given $r \leq n, m$, the (entropic) low rank optimal transport problem Scetbon et al. (2021); Scetbon & Cuturi (2022) is defined as

$$LrOT_r(\mu, \nu) := \min_{\gamma \in U(p^1, p^2; r)} \sum_{i,j} c(x_i, y_j) \Gamma_{i,j} - \epsilon H(\Gamma). \tag{5}$$

where $U(p^1, p^2; r) := U(p^1, p^2) \cap \{\Gamma \in \mathbb{R}_+^{n \times m} : \text{rk}_+(\Gamma) \leq r\}$.

From Theorem 3.2 in Cohen & Rothblum (1993) (Also see (5) in Scetbon & Cuturi (2022)), we have

$$U(p^1, p^2; r) := U(p^1, p^2) \cap \{\Gamma \in \mathbb{R}_+^{n \times m} : \text{rk}_+(\Gamma) \leq r\}$$
$$= \{\Gamma = (\Gamma^1)^\top \text{diag}(1/\sigma) \Gamma^2 : \Gamma^1 \in U(p^0, p^1), \Gamma^2 \in U(p^0, p^2), p^0 \in \Delta_r^+\}.$$

Thus, the low-rank optimal transport equation 5 becomes

$$LrOT_r(\mu, \nu) = \min_{p^0 \in \Delta_r^+} \min_{\substack{\Gamma = (\Gamma^1)^\top \text{diag}(1/\sigma) \Gamma^2: \\ \Gamma^1 \in U(p^0, p^1), \Gamma^2 \in U(p^0, p^2)}} \sum_{i,j} c(x_i, y_j) \Gamma_{i,j} - \epsilon H(\Gamma) \tag{6}$$

**Main theoretical result between LOT and LrOT** With a little abuse of notations, we use $\sigma$ to denote both the reference measure (pmf and locations) and its pmf. And define the LOT distance introduced in the main text:

$$LOT(\mu, \nu; \sigma) = \sum_{i,j} c(x_i, y_j) \Gamma_{i,j}, \Gamma = (\Gamma^1)^\top \text{diag}(1/\sigma) \Gamma^2 + \epsilon(\Gamma) \tag{7}$$

$\Gamma^1$ is optimal to $OT_\epsilon(\sigma, \mu), \Gamma^2$ is optimal to $OT_\epsilon(\sigma, \nu)$.

Let $\mathcal{P}_r(\mathbb{R}^d)$ denote the set of all discrete measure whose size is $r$, we consider the following optimal LOT problem:

$$LOT_r(\mu, \nu) := \inf_{\sigma \in \mathcal{P}_r(\mathbb{R}^D)} LOT(\mu, \nu; \sigma) \tag{8}$$

We demonstrate the relation between LOT and low-rank OT via the following proposition:

**Lemma B.1.** *In the finite discrete measure setting, we have*

$$LrOT_r(\mu, \nu) \leq LOT_r(\mu, \nu), \forall \epsilon \geq 0$$

*Proof.* For each $\sigma$, from the definition of $LOT(\mu, \nu; \sigma)$, we the $\gamma^1, \gamma^2$ in the definition of LOT equation 7 satisfy $\gamma^1 \in \Gamma(p^0, p^1), \gamma^1 \in \Gamma(p^1, p^2)$ and $p^0 \in \Delta_r^+$ by the definition of $\sigma$. Thus,

$$LrOT_r(\mu, \nu) \leq LOT(\mu, \nu; \sigma).$$

Taking the infimum over all $\sigma$ for both sides, we obtain

$$LrOT_r(\mu, \nu) \leq LOT_r(\mu, \nu),$$

and complete the proof. $\square$

## C    LOTFORMER AND SOFT CLUSTERING.

LOTFormer effectively performs a soft clustering of queries and keys via the pivot measure, establishing correspondences between these clusters. Attention (message passing) is then mediated through the pivot.

Formally, given the optimal plan $\gamma^1$ for $OT(\mu, \sigma)$, for each pivot $s_i, i \in [1 : r]$, define the sub-probability measure

$$C_i^\mu = \gamma^1(\cdot, s_i) = \sum_{j=1}^n \gamma_{j,i}^1 \delta_{x_j}.$$

The collection $\{C_1^\mu, \ldots, C_r^\mu\}$ forms a soft clustering of queries. Similarly, from the optimal plan $\gamma^2$ for $OT(\sigma, \nu)$, we obtain

$$C_i^\nu = \gamma^2(s_i, \cdot) = \sum_{k=1}^m \gamma_{i,k}^2 \, \delta_{y_k},$$

yielding a soft clustering of keys. Crucially, $C_i^\mu$ and $C_i^\nu$ are coupled through their common pivot $s_i$.

In fact, the glued coupling can be written as a sum of outer products of the soft clusters. Specifically,

$$\Gamma = (\Gamma^{(1)})^\top \operatorname{Diag}(\sigma)^{-1} \Gamma^{(2)} = \sum_{i=1}^r \frac{1}{\sigma_i} \, C_i^\mu \otimes C_i^\nu,$$

where $C_i^\mu$ and $C_i^\nu$ are the soft clusters of queries and keys induced by the pivot $s_i$. This decomposition shows that attention is mediated through correspondences between query and key clusters.

The soft-clustering perspective connects our framework to a broad line of work that leverages clustering to define attention. For example, the *Routing Transformer* employs k-means clustering of queries for localized attention (Roy et al., 2021), while the *Reformer* uses LSH-based bucketing of queries and keys (Kitaev et al., 2020).

## D  IMPLEMENTATION DETAILS

Table 7: Hyperparameters for Long Range Arena

| Task | Learning Rate | Sinkhorn Eps | Reference Mass Temp |
|------|---------------|--------------|---------------------|
| Text | $1e-4$ | 5.0 | 0.05 |
| ListOps | $1e-3$ | 0.05 | 8.0 |
| Retrieval | $3e-4$ | 0.1 | 0.1 |
| Pathfinder | $2e-4$ | 0.05 | 0.5 |
| Image | $2e-3$ | 0.05 | 4.0 |

### D.1  LONG RANGE ARENA

We implement our method inside of the Skyformer Chen et al. (2021) codebase and, unless otherwise specified, adopt the same hyperparameters and training schedule as them. For all tasks, we perform learning rate warmup for $5,000$. We use a cosine learning rate scheduler with a minimum learning rate factor of $0.1$. For each task, we sweep learning rate, sinkhorn epsilon, and reference mass temperature. We perform this sweep without a DWC. We select the best hyperparameters based on performance on validation data. For all tasks, we fix number of sinkhorn iterations to 5. We provide the hyperparameters found for each dataset in Table 7. When adding the DWC, we set the kernel size to 3 and utilize a 1d convolution for Text, Retrieval, and ListOps tasks and 2d convolution for Text and Pathfinder. All other hyperparameters are kept fixed.

### D.2  IMAGENET1K

Table 8: Hyperparameters for ImageNet-1K Classification

| Dataset | Optimizer | Learning Rate (bs=1024) | Weight Decay | Epochs | Warmup |
|---------|-----------|-------------------------|--------------|--------|--------|
| ImageNet-1K | AdamW | $3 \times 10^{-4}$ | $5 \times 10^{-2}$ | 400 | 10 (Cosine) |

We implement our method on top of the official Deit-Tiny training framework and, unless otherwise stated, keep the same training setup. All ImageNet-1K models are trained for 400 epochs using the AdamW optimizer with linear warmup for the first 10 epochs. The base learning rate is $3e-4$ for a global batch size of 1024, and we use weight decay $5e-2$.

### D.3 Neural Machine Translation

For our neural machine translation experiments, we adopt the Transformer model from `fairseq` along with its DiffTransformer counterpart, training both from scratch for 25 epochs. We then fine-tune them alongside other baselines for an additional 10 epochs on the IWSLT'14 dataset[1].

---

[1]`https://github.com/pytorch/fairseq/blob/main/examples/translation/README.md`

