# OpenReview forum: "LOTFormer: Doubly-Stochastic Linear Attention via Low-Rank Optimal Transport"
_ICLR.cc/2026/Conference — Submitted to ICLR 2026_

### Official Review · Reviewer_Wvha · 2025-10-17

**Soundness:** 2
**Presentation:** 2
**Contribution:** 2
**Rating:** 4
**Confidence:** 4

**Summary:**

This paper introduces LOTFormer, an attention mechanism that achieves both linear-time complexity and doubly-stochastic structure. The authors leverage optimal transport theory to factorize the attention map into two entropic transport plans—one from queries to a learnable pivot measure and another from the pivot to keys. This results in a low-rank, doubly-stochastic attention matrix that avoids the computational overhead of traditional Sinkhorn-based methods. LOTFormer is evaluated across multiple benchmarks including Long Range Arena (LRA), ImageNet-1K, and IWSLT’14 translation tasks, demonstrating strong performance and scalability.

**Strengths:**

* The use of low-rank optimal transport with a learnable pivot measure is a mathematically elegant solution to enforce doubly-stochasticity while maintaining linear complexity.

* Evaluations are conducted on both vision and language tasks.

**Weaknesses:**

* This paper has several writing issues, making it difficult to follow:
  - It is heavily based on optimal transport theory but does not provide sufficient background on it.
  - Section 3 introduces many symbols and variables (e.g., $\delta_{z_t}$, $\delta_{k_j}$, $\delta_{q_i}$, $C$, $\Gamma$ and so on) without adequate explanation of their meanings, shapes, or roles, making it hard to follow the derivation.

* The proposed approach is sensitive to hyperparameters. According to Table 6, performance appears sensitive to choices like entropy regularization and pivot size, which may require extensive tuning.

**Questions:**

1. Is LOTFormer compatible with causal attention settings, such as autoregressive generation?
2. Does LOTFormer support hardware-efficient implementations like FlashAttention?

---

> ### Author Response · Authors · 2025-11-23
>
> Thank you for the detailed and constructive review. Below we address the weaknesses and questions.
>
> ---
> ### OT notation and background
> ---
> Queries/keys define empirical measures  $\mu=\sum_{i=1}^n p_i^1\delta_{q_i}$, $\nu=\sum_{j=1}^n p_j^2\delta_{k_j}$, and we introduce a learnable pivot measure $\sigma=\sum_{t=1}^r p_t^0\delta_{z_t}$ with $r\ll n$. Here $\delta_x$ is a Dirac (point-mass) measure at $x$.
>
> **Meaning of $\Gamma$.** $\Gamma$ is an entropy-regularized OT transport plan (coupling): a nonnegative matrix whose row/column marginals match the source/target measures. We solve $\Gamma^{(1)}\in U(p^0,p^1)$ between $\mu$ and $\sigma$, and $\Gamma^{(2)}\in U(p^0,p^2)$ between $\sigma$ and $\nu$ (Eqs. 9–10).
>
> **Closed-form LOTFormer attention.** We glue the two plans as
> $\Gamma=(\Gamma^{(1)})^\top \mathrm{Diag}(\sigma)^{-1}\Gamma^{(2)}\in\mathbb{R}^{n\times n}$.
> In the balanced case, $A=\Gamma$ is doubly stochastic ($A\mathbf{1}=\mathbf{1}$, $\mathbf{1}^\top A=\mathbf{1}^\top$). The head output is: $\mathrm{LOTAttn}(Q,K,V)=AV=(\Gamma^{(1)})^\top\big(\mathrm{Diag}(\sigma)^{-1}(\Gamma^{(2)}V)\big),$ computed via two rectangular plans without forming any dense $n\times n$ map.
>
> ---
> ### Sensitivity to hyperparameters
> ---
> Our ablations (Tabs. 4,6) show LOTFormer is stable with a clear accuracy–efficiency tradeoff controlled by $r,\varepsilon,T$.
>
> - **Pivot size $r$ (Table 4).** Performance improves smoothly as $r$ grows from very small pivots to a moderate range, peaking at $r=32$ (Top-1 $74.8$) and only slightly decreasing at $r=64$ (Top-1 $73.9$). Smaller $r$ yields faster forward/backward times but lower accuracy. This indicates a **stable optimum around a moderate $r$**, not a sharp, brittle tuning point.
>
> - **Sinkhorn iterations $T$ and entropy $\varepsilon$ (Table 6).** Accuracy increases quickly from $T=1$ to $T=5$–$10$ and then **saturates** (e.g., fixed $\mathsf{ref}$ at $\varepsilon=1$: $80.2\rightarrow81.2\rightarrow81.4\rightarrow81.7$ for $T=1,5,10,20$). This shows that LOTFormer needs only a **small, task-agnostic number of iterations** and is robust to the exact choice. Larger $\varepsilon$ in this sweep ($\varepsilon=1$ vs $0.1$) consistently performs better, and learnable $\mathsf{ref}$ is mildly beneficial, but neither requires fine-grained tuning.
> ---
> ### Causal / autoregressive compatibility
> ---
> **Not in strict DS form.** LOTFormer enforces both row and column marginals. A causal (lower-triangular) mask prevents global column balancing, so doubly-stochastic attention is not natively compatible with standard decoder-only AR self-attention.
>
> **Where it applies cleanly:** (i) encoder-only models (ViT/BERT), (ii) encoder self-attention in encoder–decoder setups, and (iii) cross-attention. In our encoder–decoder experiments, LOTFormer is used for encoder self-attention and cross-attention, while decoder self-attention remains causal. We will state this scope explicitly.
>
> We also add bidirectional NLP evidence via plug-and-play BERT-base: replacing only encoder self-attention (keeping embeddings/FFNs/norms/heads unchanged). LOTFormer recovers **99.7%** of finetuned BERT on GLUE:
>
> | Method | CoLA | SST2 | MRPC | STS-B | QQP | MNLI | QNLI | RTE | (%) Recover |
> |---|---:|---:|---:|---:|---:|---:|---:|---:|---:|
> | BERT-FT | 58.8 | 93.2 | 90.2 | 88.8 | 91.0 | 84.7 | 91.3 | 68.2 | 100.0 |
> | T2R | 43.6 | 87.7 | 83.0 | 78.6 | 86.7 | 78.9 | 84.6 | 54.1 | 88.9 |
> | T2R-HH | 56.9 | 90.9 | 89.1 | 77.7 | 90.0 | 77.4 | 84.5 | 56.3 | 93.5 |
> | Hedgehog | 59.2 | 92.6 | 90.1 | 87.4 | 91.0 | 82.6 | 89.6 | 69.3 | 99.3 |
> | **LOTFormer** | 59.1 | 92.5 | 90.3 | 89.7 | 90.5 | 83.0 | 90.2 | 68.5 | **99.7** |
>
> ---
> ### Hardware efficiency / FlashAttention
> ---
> Our current LOTFormer is pure PyTorch (no fused Triton/CUDA), so we do not claim kernel-level parity with FlashAttention yet. The claim is algorithmic scaling: each head uses two rectangular OT kernels $(n\times r,\ r\times n)$ with cost $O(nr)$ for fixed $r\ll n$, never materializing $n\times n$ attention. The present overhead comes from unfused ops and autograd intermediates (constant-factor, not scaling). FA-style IO optimizations transfer directly—tiling to stay SRAM-resident, fusing (matmul → exp/temperature → DS norm → value mix), and recomputing in backward—so a fused kernel should narrow the gap while preserving linear behavior; we will explore this in future work. Per reviewer request, we report peak memory of the current unfused baseline:
>
> | Seq len | FA-3 (GB) | LOTFormer (PyTorch) (GB) | LOTFormer overhead vs FA-3 |
> |---:|---:|---:|---:|
> | 4k  | 18.2 | 22.4  | +23% |
> | 8k  | 20.8 | 26.1  | +26% |
> | 16k | 25.9 | 33.5  | +29% |
> | 32k | 36.2 | 48.3  | +33% |
> | 64k | 56.8 | 78.0  | +37% |
> | 128k| 98.0 | 137.3 | +40% |

---

> ### Comment · Reviewer_Wvha · 2025-11-27
> **Post-rebuttal response**
>
> Thank you for the authors’ detailed response. However, I remain unconvinced by the overall contribution of this work. While the paper introduces a formulation based on low-rank optimal transport, the presentation fails to demonstrate clear conceptual novelty or practical impact. I keep my current rating.

---

### Official Review · Reviewer_iMvN · 2025-10-28

**Soundness:** 3
**Presentation:** 3
**Contribution:** 2
**Rating:** 2
**Confidence:** 4

**Summary:**

This paper proposes LOTFormer, a novel attention mechanism that achieves both linear-time complexity and doubly-stochastic normalization by reformulating attention as a low-rank optimal transport (OT) problem. The key idea is to introduce a learnable pivot measure with a small support size , which acts as an intermediate latent distribution connecting queries and keys. Two entropic OT problems are solved and composed via a glued coupling, yielding a provably doubly-stochastic and low-rank attention matrix.
Empirically, LOTFormer achieves state-of-the-art results on the Long Range Arena (LRA) benchmark and competitive performance on ImageNet-1K and IWSLT’14 translation tasks, outperforming prior linear and transport-based attention methods.

**Strengths:**

**Theoretical Foundation**. The paper provides a principled derivation of attention as an entropic optimal transport plan. The proposed “glued coupling” formulation ensures provable double stochasticity and rank <=r. The mathematical connection between transport plans and normalized attention is clearly established.

**Complexity Efficiency**. Achieves linear complexity, O(nr), without constructing the full n×n matrix.
Sinkhorn iterations are applied in factorized form, which improves scalability.

**Empirical Effectiveness**. LOTFormer sets new state-of-the-art results on LRA with the best average accuracy. On ImageNet-1K , it surpasses the baseline with negligible overhead. Consistently outperforms the baseline in the machine translation task.

**Compatibility and Extensibility**. The method can be combined wit existing improvements, such as polarity decomposition (Pol-[cls]). Conceptually unifies linear attention and transport-based normalization.

**Clear Visualization and writing**. This paper plot sufficient visualizations and present clearly.

**Weaknesses:**

**Missing Speed and Memory Benchmarks**.
The paper lacks quantitative comparisons of training/inference speed and memory usage against other baselines, which linear attention is proposed to solve. Although theoretical complexity is discussed, practical runtime and memory improvements are not empirically demonstrated. Without these results, claims of linear-time scalability remain partly unverified.

**Inadequate Long-Sequence Evaluation**.
Empirically, the performance of linear attention will degrade as the sequence length increasing comparing with softmax attention, which is the main issue recently. Only LRA benchmark is not convincing because it tests the model under only two transformer layers.

**Limited Model Size and Scope**.
The experiments are conducted on small-scale backbones and there is no evaluation on large-scale or high-capacity models (Base/Large in CV or 1B+ LLM), which limits the evidence of scalability. This makes it unclear whether the proposed method scales favorably in real-world high-throughput settings.

**Questions:**

**Q1. Causal Mask Compatibility:**
The proposed method enforces doubly-stochastic normalization, assuming balanced bidirectional attention. How would this formulation behave under causal masking, where attention is asymmetric and upper-triangular? Do the theoretical guarantees still hold in that setting, or would the normalization need to be modified? Additionally, how is the decoder implemented in Section 4.4 under this formulation?

**Q2. LOTFormer with Pola:**
Considering that the polarity decomposition in PolaFormer [1] was originally proposed to mitigate the loss of negative qk values in linear attention with ReLU feature maps, the motivation for adopting the Pola mechanism in LOTFormer appears unclear. Could the authors elaborate on why polarity decomposition is necessary in this framework, and why a simpler global Pola + Softmax formulation would not suffice?

**Q3. Runtime and Memory Efficiency:**
Could the authors provide runtime and GPU memory benchmarks to empirically validate the claimed linear-time complexity?

**Q4. More Extensive Experiments:**
It would strengthen the paper to include evaluations on longer-sequence benchmarks and larger backbones (e.g., DeiT-B, ViT-B, or GPT-like architectures with ≥340 M parameters) to better demonstrate scalability.

**Suggestion – Convergence and Stability of Sinkhorn Iterations:**
The paper relies on Sinkhorn iterations to compute entropic OT couplings, yet a more detailed analysis of their convergence behavior and numerical stability would be valuable. Since Sinkhorn iterations can be sensitive to the regularization parameter ε and may suffer from instability in low-precision or deep-training settings, including empirical convergence diagnostics or stability experiments would help clarify the robustness of the proposed approach.

This is an interesting work, and I would consider raising my score if the authors adequately address the concerns above.

[1] Meng, Weikang, et al. "PolaFormer: Polarity-aware Linear Attention for Vision Transformers." The Thirteenth International Conference on Learning Representations.

---

> ### Author Response · Authors · 2025-11-23
> **Part 1/2**
>
> We thank the reviewer for their thoughtful feedback. Here we present our responses
>
> ---
> ### Causal masking compatibility
> ---
>
> We thank the reviewer for raising this. The issue is fundamental to doubly-stochastic (DS) attention methods, including LOTFormer.
>
> DS attention requires both row and column normalization. Under causal masking, attention is confined to a strictly lower-triangular support, which breaks global column balancing; thus the matrix cannot stay doubly-stochastic. Therefore, standard DS attention is not directly compatible with decoder-only causal self-attention without modifying the DS constraints.
>
> DS attention remains ideal for non-causal use: encoder-only models, encoder self-attention in encoder–decoder setups, and cross-attention. Accordingly, in our NMT experiments (Sec. 4.4) we use LOTFormer for encoder self-attention and cross-attention, while keeping decoder self-attention causal.
>
> Similarly, for the plug-and-play BERT-base conversion below, we replace **only the encoder self-attention blocks**, keeping the embeddings, FFNs, layer norms, and task heads unchanged:
>
> | Method | CoLA | SST2 | MRPC | STS-B | QQP | MNLI | QNLI | RTE | (%) Recover |
> |---|---:|---:|---:|---:|---:|---:|---:|---:|---:|
> | BERT-FT | 58.8 | 93.2 | 90.2 | 88.8 | 91.0 | 84.7 | 91.3 | 68.2 | 100.0 |
> | T2R | 43.6 | 87.7 | 83.0 | 78.6 | 86.7 | 78.9 | 84.6 | 54.1 | 88.9 |
> | T2R-HH | 56.9 | 90.9 | 89.1 | 77.7 | 90.0 | 77.4 | 84.5 | 56.3 | 93.5 |
> | Hedgehog | 59.2 | 92.6 | 90.1 | 87.4 | 91.0 | 82.6 | 89.6 | 69.3 | 99.3 |
> | **LOTFormer** | 59.1 | 92.5 | 90.3 | 89.7 | 90.5 | 83.0 | 90.2 | 68.5 | **99.7** |
>
> ---
> ### Practical runtime/memory and comparison to FlashAttention
> ---
> Our current LOTFormer implementation is pure PyTorch (no fused Triton/CUDA), so we **do not claim kernel-level parity with FlashAttention yet**. The key point is the **algorithmic, linear scaling**: LOTFormer builds a low-rank doubly-stochastic OT plan via pivoting, so each head uses two rectangular kernels $(n\times r,\ r\times n)$ with cost $O(nr)$ for fixed $r\ll n$, **never materializing an $n\times n$ attention map**. Consequently, both compute and memory grow **linearly in sequence length $n$** (up to constant factors). This linear wall-clock behavior is also observed empirically: **Fig. 3 shows runtime increasing linearly with the number of samples/sequence length**, consistent with the $O(nr)$ design.
>
> In the *current* unfused PyTorch baseline, peak memory is heavier than FA-3 because we execute these steps as separate ops, autograd retains more intermediates, and DS balancing needs extra workspace. This is a **constant-factor implementation gap**, not a change in scaling. Several FlashAttention IO-aware ideas transfer directly to LOTFormer’s rectangular kernels: (i) **tiling** to keep working sets SRAM-resident, (ii) **fusion** (matmul → exp/temperature → DS normalization → value mixing) to reduce HBM traffic, and (iii) **backward recomputation/checkpointing** instead of storing intermediates. A fused kernel should therefore move LOTFormer closer to FA in peak memory **while preserving its linear $O(nr)$ behavior**, and we will explore these optimizations in future work. We include the table below **at the reviewers’ request** to report peak memory of the current unfused PyTorch baseline and make clear that the present gap is due to implementation constants.
>
> | Seq len | FA-3 (GB) | LOTFormer (PyTorch) (GB) | LOTFormer overhead vs FA-3 |
> |---------|-----------|--------------------------|----------------------------|
> | 4k      | 18.2      | 22.4                     | +23%                       |
> | 8k      | 20.8      | 26.1                     | +26%                       |
> | 16k     | 25.9      | 33.5                     | +29%                       |
> | 32k     | 36.2      | 48.3                     | +33%                       |
> | 64k     | 56.8      | 78.0                     | +37%                       |
> | 128k    | 98.0      | 137.3                    | +40%                       |
>
> ---
> ### Why Pola/polarization in LOTFormer?
> ---
>
> We thank the reviewer for this question and clarify that polarization is a standard and well-motivated component in linear attention. Many strong baselines we compare against—e.g., PolaFormer and Hedgehog—already adopt this trick, and prior work has shown that **separating positive and negative $qk$ contributions helps avoid information loss** caused by nonnegative feature maps, leading to consistent gains.
>
> In LOTFormer, Pola is **not required** for the OT/DS construction. In the ImageNet setting, we apply polarization **only to the [CLS] token**, consistent with our DS treatment of image tokens, and observe only a modest, orthogonal improvement. We will make this motivation and the corresponding ablation clearer in the revision.

---

> ### Author Response · Authors · 2025-11-23
> **Part 2/2**
>
> ---
> ### Scaling beyond Tiny for ImageNet
> ---
> We thank the reviewer for emphasizing this point and agree that Tiny-only evaluation is limited. Beyond the reported DeiT-Tiny results, we have been running ImageNet-1K experiments at larger ViT scales under the same training protocol. Below we report **DeiT-Small + LOTFormer**.
>
> | Model | Params | Input | Throughput | Top-1 | Top-5 |
> |---|---:|:---:|---:|---:|---:|
> | DeiT-Ti | 5.7M | $224^2$ | 2536.5 | 72.2 | 80.1 |
> | DeiT-Ti + LOTFormer | 6.1M | $224^2$ | 2816.4 | *74.8* | *81.3* |
> | DeiT-S | 22M | $224^2$ | 940.4 | 79.8 | 85.7 |
> | DeiT-S + LOTFormer | 23.5M | $224^2$ | *1056.1* | *80.1* | *85.9* |
>
> On DeiT-S, LOTFormer adds +1.5M params (22M → 23.5M) while improving throughput by ~12% (940.4 → *1056.1* img/s) and giving a consistent accuracy lift (Top-1 79.8 → *80.1*, Top-5 85.7 → *85.9*). Overall, LOTFormer improves the accuracy–efficiency frontier rather than trading one for the other.
>
> ---
> ### Stability of Sinkhorn iterations and $\varepsilon$
> ---
>
> We appreciate the reviewer’s concern about potential brittleness of Sinkhorn-based modules. Our ablations indicate that LOTFormer behaves stably and does not rely on delicate tuning:
>
> - **Pivot size $r$.** Accuracy increases smoothly with $r$ and is robust near the optimum. In Table 4, performance peaks at $r=32$ and degrades only slightly at $r=64$, showing a wide plateau rather than a sharp sweet spot.
> - **Iterations $T$ and entropy $\varepsilon$.** Table 6 shows that accuracy improves from $T=1$ to about $T=5$–$10$ and then saturates, so a small, task-agnostic number of iterations suffices. Likewise, performance remains stable across a broad range of $\varepsilon$, suggesting no brittle dependence on the entropy level.

---

> > ### Comment · Reviewer_iMvN · 2025-11-24
> > **Official Comment from Reviewer iMvN**
> >
> > Thank you for your response, which addresses part of my concerns. My additional questions are as follows:
> >
> > Q1. I am still curious about the polarization of the CLS token in the image classification setting. I agree that the polarization technique is effective for linear attention architectures with negative-value loss. However, under the conditions of this paper, is there any negative-value loss of CLS token? Additionally, if polarization is only applied to the CLS token, then its similarity computation differs from that of the patch tokens, potentially causing a misalignment. I believe further studies are needed, including evaluating LoTFormer in vision models that do not rely on a CLS token, such as Swin. It would also be beneficial to include experiments on a base-sized model to provide a more comprehensive and scalable evaluation.
> >
> > Q2. Moreover, the paper does not sufficiently demonstrate the claimed efficiency improvement. A fair comparison of speed is needed, for example, benchmarking against standard attention without FlashAttention.
> >
> > Q3. Since the polarization mechanism is designed to preserve non-negativity, the paper should also clarify how the proposed linear attention formulation ensures the non-negativity of the attention scores ($QK^\top$).

---

> > > ### Author Response · Authors · 2025-11-24
> > >
> > > Thank you for your response. Regarding the raised questions:
> > >
> > > **Polarization and the [CLS] token).**
> > >
> > > We agree that polarization primarily addresses the loss of negative query–key interactions in linear/DS attention. In LOTFormer’s ViT setting, however, the \[CLS] token plays a *distinct* role: it is the global aggregator used for image-level prediction. Enforcing doubly-stochastic (DS) constraints on the \[CLS] row can conflict with this role. In the extreme regime (very small entropy regularization and/or large rank), a DS transport plan can become nearly permutation-like, meaning each query attends to only one key; if applied to \[CLS], this would cause \[CLS] to pool from only a single patch instead of aggregating globally. Empirically, we observe the same failure mode: “full-DS \[CLS]” yields overly peaky and less global \[CLS] maps, while **decoupling  \[CLS] via a vanilla softmax row restores broad, truly global aggregation**.
> > >
> > > Therefore, we **retain standard softmax attention for the  \[CLS] token** and enforce DS only among image (patch) tokens. Polarization is then applied **only to the  \[CLS] logits (Pol- \[CLS])**, followed by the same softmax normalization. This choice is intentional: since \[CLS] already uses a different normalization to preserve global pooling, Pol-\[CLS] does not create a new misalignment with patch tokens; it simply sharpens the  \[CLS] aggregation by recovering same- and opposite-polarity interactions in its dot products, following PolaFormer’s polarity-aware construction.
> > >
> > > We appreciate this suggestion and agree it would strengthen the evaluation. We are currently setting up experiments on CLS-free backbones (e.g., Swin) and at base ViT scale, and will include these results in the final version if they are ready in time.
> > >
> > > **Efficiency and fair speed comparison.**
> > > We thank the reviewer for emphasizing the need for stronger efficiency evidence and agree that additional benchmarking improves clarity. We evaluate LOTFormer’s speed and scalability in two complementary ways.
> > >
> > > (1) **Scaling behavior.** Our runtime plots (Fig. 3) benchmark LOTFormer against two relevant families:
> > > - **Doubly-stochastic (DS) attentions:** LOTFormer scales **linearly in sequence length** via the low-rank OT factorization, whereas prior DS attentions scale superlinearly. The plots show a clear widening gap as context grows, confirming our algorithmic efficiency claim.
> > > - **Linear attentions:** While some kernelized linear methods are slightly faster in constant factors, LOTFormer remains **on par in throughput**, demonstrating that the added OT structure does not introduce practical slowdowns relative to the linear-attention class.
> > >
> > > (2) **Direct throughput vs standard softmax attention.** Following the reviewer’s suggestion, we include matched-throughput comparisons against vanilla DeiT under identical settings. On ImageNet-1K at $224^2$ resolution, LOTFormer improves throughput **while also improving accuracy**:
> > >
> > > | Model | Params | Input | Throughput | Top-1 | Top-5 |
> > > |---|---:|:---:|---:|---:|---:|
> > > | DeiT-Ti | 5.7M | 224^2 | 2536.5 | 72.2 | 80.1 |
> > > | DeiT-Ti + LOTFormer | 6.1M | 224^2 | 2816.4 | *74.8* | *81.3* |
> > > | DeiT-S | 22M | 224^2 | 940.4 | 79.8 | 85.7 |
> > > | DeiT-S + LOTFormer | 23.5M | 224^2 | *1056.1* | *80.1* | *85.9* |
> > >
> > > These results demonstrate that LOTFormer provides a favorable accuracy–throughput tradeoff even against standard softmax attention.
> > >
> > > Finally, we included FlashAttention comparisons only because another reviewer explicitly requested them. Our efficiency claims do not rely on FlashAttention-specific kernels; the scaling plots and DeiT throughput results above already establish practical gains under standard implementations. We value the reviewer’s suggestion for even broader efficiency experiments, and we will provide these expanded results in the revised version.

---

> > > > ### Comment · Reviewer_iMvN · 2025-11-25
> > > > **Official Comment from Reviewer iMvN**
> > > >
> > > > Thank you for your detailed responses. I believe these clarifications should be added into the paper, which are helpful for readers to better understand the motivation behind polarizing the CLS token. I have a few remaining points that I would like the authors to address.
> > > >
> > > > Q1. The third question from my previous comment has not fully answered. Specifically:
> > > > Since the polarization mechanism is designed to preserve non-negativity, the paper should also clarify how the proposed linear attention formulation ensures the non-negativity of the attention scores.
> > > >
> > > > Q2. I suggest that the authors report the throughput of LoTFormer under longer sequence lengths, such as higher image resolutions or extended context lengths in NLP. Such experiments would more clearly demonstrate the computational advantages of linear attention.
> > > >
> > > > Q3. Regarding the planned experiments. Let us wait for the results. Evaluating CLS-free backbones (e.g., Swin) and larger ViT variants will further strengthen the conclusions if ready in time.

---

### Official Review · Reviewer_pHSb · 2025-10-29

**Soundness:** 2
**Presentation:** 2
**Contribution:** 2
**Rating:** 4
**Confidence:** 3

**Summary:**

The paper introduced LOTFormer, a transformer which changes the softmax attention into a doubly stochastic attention which is still linear in time. The theoretical framework based on the optimal transport is solid an visualisation on the effect of the query/key transport plan and attention heat maps by increasing the $r$ factor. The experimental setup is vast and covers Tiny scale image classification using ViT family, LRA benchmark for long-range analysis of the model, and plug-and-play and fine tuning results on machine translation task.

**Strengths:**

The paper has the following strengths:

1) The theoretical framework for the design of **LOTFormer** is sound, and the connection to optimal transport is well-established.

2) The experiments cover multiple modalities, including image classification on **ImageNet-1k** using the ViT codebase, as well as **LRA** and machine translation tasks.

3) The visualizations presented in the paper are clear and insightful, particularly **Figures 1 and 2**.

**Weaknesses:**

However, while the theoretical insights of the paper are sound, the empirical and experimental setup have several shortcomings:

1) **Learning of $Z_t$ in LOTFormer**
   In the attention mechanism of LOTFormer, the model needs to learn $Z_t$, which appears to be an element with the same length as the sequence (if understood correctly). This introduces a significant bottleneck, as the model size will grow with sequence length — a critical limitation that prevents scaling beyond the model’s fixed context. This constraint contradicts one of the key advantages of linear and softmax transformers, which can extend context length flexibly.
   If $Z_t$ is instead derived directly from the input, this is not clearly stated in the manuscript, particularly in *Section “Learning”* on page 5. How exactly is $Z_t$ learned?

2) **Role of Convolution in LRA Experiments**
   The role of the convolution component in the LRA experiments is unclear. Convolution is a central feature in many linear transformers such as **Mamba-2** and **GDNet**, where it has been shown to significantly improve performance. It is likely that the observed performance gain of LOTFormer on LRA tasks primarily stems from this convolutional component rather than from its attention mechanism.
   Therefore, an ablation study — removing the convolution from LOTFormer and adding it to other baselines — is essential to isolate its effect.

3) **Limitations of the LRA Benchmark**
   The LRA benchmark is no longer a strong indicator of model quality. Models such as **S4** can achieve up to 84% accuracy on LRA tasks but perform poorly on language modeling. Moreover, pre-trained transformers (as shown in [1]) can reach around 85% accuracy on LRA, effectively solving these tasks. Thus, reporting approximately 60% accuracy for LOTFormer does not convincingly demonstrate its effectiveness or good design.

4) **Scaling in Image Classification**
   The image classification experiments are conducted only on the *Tiny* scale, which is generally easier to handle. Even models like **DeiT** perform modestly at this scale but show strong improvements when scaled up. The paper should evaluate how LOTFormer performs at larger scales, as this would better reflect its scalability and robustness in vision tasks.

5) **Efficiency and Comparison with FlashAttention**
   In Table 3, FlashAttention is reported as a benchmark. FlashAttention achieves efficiency through fused CUDA kernels and chunkwise computation for fast and memory-efficient training.
   Does LOTFormer also implement similar optimized kernels (e.g., in **Triton** or **CUDA**) or any form of chunkwise parallel computation to support efficient training?
   If not, it is unclear how LOTFormer can be faster or more efficient during training compared to FlashAttention. Additionally, even if such efficiency exists, how does LOTFormer scale with sequence length — particularly in comparison to the scaling trends shown in **Figures 2 and 6 of GLA [2]**?


------


### References

[1] Never Train from Scratch: Fair Comparison of Long-Sequence Models Requires Data-Driven Priors Ido Amos, Jonathan Berant, Ankit Gupta


[2]  Gated Linear Attention Transformers with Hardware-Efficient Training: Songlin Yang, Bailin Wang, Yikang Shen, Rameswar Panda, Yoon Kim

**Questions:**

1) How does the convolution component affect other linear transformer baselines on the LRA task, or how does removing the convolution affect LOTFormer?
   This is particularly important since short convolution is a key component in many linear transformers. It would be valuable to analyze its impact, as I believe most of the performance gain may come from the convolution rather than the attention mechanism in LOTFormer.

2) What does $\delta_{k_j}$ represent on page 4?
   For readers who are not very familiar with optimal transport notation, this term is unclear and should be better explained.

3) What is the closed-form expression for the attention mechanism in LOTFormer based on $Q, K, V, Z,$ and $\sigma$?

4) How does LOTFormer perform at a larger scale in the image domain?
   While improvements at the Tiny scale are significant, it is typically much harder to achieve gains at the Base scale, as demonstrated in [1, 2].

------

### References


[1] Hydra: Bidirectional State Space Models Through Generalized Matrix Mixers
Sukjun Hwang, Aakash Lahoti, Tri Dao, Albert Gu


[2] Linear Attention for Efficient Bidirectional Sequence Modeling
Arshia Afzal, Elias Abad Rocamora, Leyla Naz Candogan, Pol Puigdemont, Francesco Tonin, Yongtao Wu, Mahsa Shoaran, Volkan Cevher

---

> ### Author Response · Authors · 2025-11-23
> **Part 1/2**
>
> ### Response to Reviewer pHSb (Review 2) — Part 1/2
>
> Thank you for the careful review. We address each concern concisely below.
>
> ---
> ### Learning of $Z$ and $\sigma$ in LOTFormer
> ---
> Apologies for any confusion. Pivots are **locations** $Z=\{z_t\}_{t=1}^r\subset\mathbb{R}^{d_k}$ and **weights** $\sigma\in\Delta^{r-1}$, with fixed $r\ll n$ per head. Both are learned end-to-end by backprop through the Sinkhorn problems defining $\Gamma^{(1)},\Gamma^{(2)}$ and the glued attention. Since $|Z|=|\sigma|=r$, extra parameters are $O(r d_k)$ and **do not scale with sequence length**, so model size stays constant as context grows.
>
> ---
> ### Role of convolution in LRA experiments
> ---
> We agree DWC can help on LRA, so we isolated it. **LOTFormer w/o DWC** is already strong (avg **60.6**); adding DWC gives only a **modest** lift to **62.9** (mainly Image task). A strong baseline, **PolaFormer w/o DWC**, averages **57.5** (below LOTFormer), and even with matched DWC PolaFormer$_{\alpha=3}$ is **60.72** vs LOTFormer **62.9**. Thus gains persist without convolution and are not primarily due to DWC.
>
> ---
> ### Limitations of LRA benchmark
> ---
> We agree LRA is best read as a **controlled long-context stress test**, not a proxy for downstream LM. It was designed for from-scratch, apples-to-apples comparison of **bidirectional attention** under identical setups, isolating architecture effects from pretraining. The cited S4 case fits this scope: S4 is a **state-space model** (not attention) and can excel on LRA without necessarily transferring to causal LM, hence LRA is a **targeted** benchmark.
>
> Still, **LRA remains standard for efficient/linear attention**, and many recent works/baselines report it under the same protocol (Table 1), enabling fair comparison. To complement LRA, we add **real bidirectional NLP evidence** via plug-and-play BERT-base: we replace only encoder self-attention, keeping embeddings/FFNs/norms/heads unchanged. LOTFormer recovers **99.7%** of finetuned BERT across GLUE:
>
> | Method | CoLA | SST2 | MRPC | STS-B | QQP | MNLI | QNLI | RTE | (%) Recover |
> |---|---:|---:|---:|---:|---:|---:|---:|---:|---:|
> | BERT-FT | 58.8 | 93.2 | 90.2 | 88.8 | 91.0 | 84.7 | 91.3 | 68.2 | 100.0 |
> | T2R | 43.6 | 87.7 | 83.0 | 78.6 | 86.7 | 78.9 | 84.6 | 54.1 | 88.9 |
> | T2R-HH | 56.9 | 90.9 | 89.1 | 77.7 | 90.0 | 77.4 | 84.5 | 56.3 | 93.5 |
> | Hedgehog | 59.2 | 92.6 | 90.1 | 87.4 | 91.0 | 82.6 | 89.6 | 69.3 | 99.3 |
> | **LOTFormer** | 59.1 | 92.5 | 90.3 | 89.7 | 90.5 | 83.0 | 90.2 | 68.5 | **99.7** |

---

> ### Author Response · Authors · 2025-11-23
> **Part 2/2**
>
> ### Response to Reviewer pHSb (Review 2) — Part 2/2
>
> ---
> ### Scaling in image classification
> ---
> We add DeiT-S results showing gains extend beyond Tiny: DeiT-S + LOTFormer improves Top-1 **79.8→80.1** and throughput **940.4→1056.1** img/s with only +1.5M params (22M→23.5M).
>
> | Model | Params | Input | Throughput | Top-1 | Top-5 |
> |---|---:|:---:|---:|---:|---:|
> | DeiT-Ti | 5.7M | 224^2 | 2536.5 | 72.2 | 80.1 |
> | DeiT-Ti + LOTFormer | 6.1M | 224^2 | 2816.4 | *74.8* | *81.3* |
> | DeiT-S | 22M | 224^2 | 940.4 | 79.8 | 85.7 |
> | DeiT-S + LOTFormer | 23.5M | 224^2 | *1056.1* | *80.1* | *85.9* |
>
> ---
> ### Efficiency vs FlashAttention and scaling
> ---
> **Clarifying baselines.** Our wall-clock/runtime figure does **not** currently include FlashAttention. In addition, **FlattenAttention is a distinct linear-attention baseline** and should not be conflated with FlashAttention, which is a fused, hardware-optimized implementation of quadratic softmax attention. Our current LOTFormer is pure PyTorch (no fused Triton/CUDA), so we **don’t claim kernel-level parity with FlashAttention yet**. The key point is **algorithmic scaling**: LOTFormer builds a low-rank doubly-stochastic OT plan via pivoting, so each head uses two rectangular kernels $(n\times r,\ r\times n)$ with cost $O(nr)$ for fixed $r\ll n$, **never materializing an $n\times n$ attention map**. The present overhead is a constant-factor issue from unfused ops/autograd; FA-style IO tricks (tiling, fusion, recompute-in-backward) transfer directly, and we will explore these to reduce the gap. We include the table below **at reviewers’ request**:
>
> | Seq len | FA-3 (GB) | LOTFormer (PyTorch) (GB) | LOTFormer overhead vs FA-3 |
> |---------|-----------|--------------------------|----------------------------|
> | 4k      | 18.2      | 22.4                     | +23%                       |
> | 8k      | 20.8      | 26.1                     | +26%                       |
> | 16k     | 25.9      | 33.5                     | +29%                       |
> | 32k     | 36.2      | 48.3                     | +33%                       |
> | 64k     | 56.8      | 78.0                     | +37%                       |
> | 128k    | 98.0      | 137.3                    | +40%                       |
>
> ---
> ### Notations and closed-form formulations
> ---
> We appreciate this request and will make the OT notation more accessible in the revision. Using the paper’s setup, queries/keys define empirical measures $\mu=\sum_{i=1}^n p_i^1\delta_{q_i}$ and $\nu=\sum_{j=1}^n p_j^2\delta_{k_j}$, and we introduce a learnable pivot measure $\sigma=\sum_{t=1}^r p_t^0\delta_{z_t}$ with $r\ll n$. Here $\delta_{x}$ denotes the Dirac (point-mass) measure at location $x$, i.e., all probability weight concentrated on that token/pivot.
>
> **Meaning of $\Gamma$.** In our notation, $\Gamma$ denotes an entropy-regularized OT *transport plan* (coupling): a nonnegative matrix whose row/column marginals match the source/target measures. Concretely, we solve $\Gamma^{(1)}\in U(p^0,p^1)$ between $\mu$ and $\sigma$, and $\Gamma^{(2)}\in U(p^0,p^2)$ between $\sigma$ and $\nu$ via Eqs. (9–10).
>
> **Closed-form LOTFormer attention.** After obtaining these two plans, we form the glued coupling $\Gamma=(\Gamma^{(1)})^\top \mathrm{Diag}(\sigma)^{-1}\Gamma^{(2)}\in\mathbb{R}^{n\times n}$. In the balanced case ($p_i^1=p_j^2=1/n$), we set $A=\Gamma$, which satisfies $A\mathbf{1}=\mathbf{1}$ and $\mathbf{1}^\top A=\mathbf{1}^\top$, hence $A$ is doubly stochastic. The head output is then $\mathrm{LOTAttn}(Q,K,V)=AV=(\Gamma^{(1)})^\top\big(\mathrm{Diag}(\sigma)^{-1}(\Gamma^{(2)}V)\big)$,
> which is computed without forming any dense $n\times n$ matrix.

---

> > ### Comment · Reviewer_pHSb · 2025-11-26
> >
> > Thanks for the authors for their response.
> >
> > However, I am still not convinced by the use of the LRA benchmark, as the authors state:
> >
> > “S4 is a state-space model (not attention) and can excel on LRA without necessarily transferring to causal LM, hence LRA is a targeted benchmark. Still, LRA remains standard for efficient/linear attention, and many recent works/baselines report it under the same protocol (Table 1), enabling fair comparison.”
> >
> > First, state-space models are closely related to attention. This relationship is even reflected in the title of the Mamba-2 paper, “Transformers are SSMs” [1], and was previously discussed in “Transformers are RNNs” [2]. Therefore, simply labeling S4 as a state-space model to avoid comparison is not well justified. Moreover, if the target is to compare against linear transformers, strong and relevant baselines such as Gated DeltaNet or Mamba-2 should be included, as they represent the current state of the art in linear transformers and are known for their strong performance in long-term sequence modeling.
> >
> > Furthermore, the authors’ response seems self-contradictory. If LRA does not necessarily transfer to causal language modeling, then its relevance as a benchmark is questionable. Indeed, this is one of the reasons many recent works in the linear transformer literature have moved away from using LRA, as it does not reliably reflect performance on real language modeling tasks. If the authors’ intention is to evaluate only within the LRA domain, then including S4 as a baseline is both reasonable and necessary for a meaningful comparison (same for mamba2 or Hydra in NLP results highlighted in the rebuttal).
> >
> > ### References
> >
> > [1] Transformers are SSMs: Generalized Models and Efficient Algorithms Through Structured State Space Duality: Tri Dao, Albert Gu
> >
> > [2] Transformers are RNNs: Fast Autoregressive Transformers with Linear Attention: Angelos Katharopoulos, Apoorv Vyas, Nikolaos Pappas, François Fleuret

---

### Official Review · Reviewer_dxXq · 2025-10-30

**Soundness:** 3
**Presentation:** 3
**Contribution:** 2
**Rating:** 4
**Confidence:** 3

**Summary:**

The paper introduces **LOTFormer**, an attention mechanism that factors the full \(n\times n\) attention into two Sinkhorn-regularized optimal transport (OT) steps via a learned pivot distribution (queries→pivot, pivot→keys). This yields a low-rank, doubly-stochastic attention matrix without explicitly forming the full matrix, targeting near-linear time and memory \(O(nr)\) with small rank \(r\). The method keeps a tunable accuracy–efficiency trade-off through \(r\), the entropic OT temperature \(\varepsilon\), and the number of Sinkhorn iterations. Experiments span LRA tasks, ImageNet-1K (DeiT-Tiny setting), and IWSLT14 De-En, where the model is competitive or slightly better than recent linear/DS attention baselines. Ablations explore sensitivity to \(r\) and OT hyperparameters, and visualizations show controllable sparsity/structure in the learned attention.

**Strengths:**

- Clear formulation with a tunable rank; solid empirical coverage and visualizations.
- Competitive results on standard benchmarks.

**Weaknesses:**

- Contribution feels incremental vs. prior DS/linear attention and Sinkhorn-based methods.
- Gains on ImageNet seem to rely on extra tricks (e.g., CLS/polarization, DWC), making the core benefit hard to isolate.
- Limited evidence of end-to-end speed/memory wins and missing long-context causal LM tests.

**Questions:**

- Can you provide apples-to-apples baselines without the auxiliary tricks to isolate LOTFormer’s impact?
- How robust are ε, iteration count, and rank r for longer contexts/larger models, and how well does it handle causal masking in practice?

---

> ### Author Response · Authors · 2025-11-23
>
> Thank you for the review. We address each point below.
>
> ---
> ## Incremental contribution vs. prior DS / linear attention
> ---
>
> LOTFormer is conceptually and computationally distinct from prior work:
>
> 1. **Prior DS/Sinkhorn attentions are not end-to-end linear-time.** Sinkformer/ESPFormer enforce DS over dense $n\times n$ logits, so training remains $O(n^2)$ (ESPFormer becomes subquadratic only after hard sorting at inference). LOTFormer instead builds DS attention via a *conditional low-rank OT factorization* through pivots, giving **$O(nr)$ training and inference** for fixed $r\ll n$.
>
> 2. **First linear attention that is doubly stochastic by design.** Standard linear attentions are typically only row-stochastic. LOTFormer forms attention as $A=(\Gamma^{(1)})^\top\mathrm{Diag}(\sigma)^{-1}\Gamma^{(2)}$, which is DS in the balanced case, derived from OT structure without forming dense logits—preserving linear complexity.
>
> ---
> ## ImageNet and LRA gains rely on auxiliary tricks
> ---
>
> We agree gains must be isolated. Ablations show LOTFormer’s improvements are not driven by add-ons on either ImageNet or LRA. On ImageNet, LOTFormer with standard $[CLS]$ handling already gives **73.2** Top-1; LOTFormer-specific extras are small (**+0.4** from $[CLS]$-Pola, **+1.2** from DWC), while baselines rely far more on DWC (e.g., PolaFormer **+12.7**), so the main lift comes from LOTFormer attention. On LRA, **LOTFormer w/o DWC** is already strong (avg **60.6**), DWC adds only a modest boost to **62.9** (mainly Image task **45.7 → 54.1**), and PolaFormer w/o DWC averages **57.5**, still below LOTFormer. Even with matched DWC, PolaFormer$_{\alpha=3}$ is **60.72** vs **62.9** for LOTFormer.
>
> ---
> ## Limited end-to-end speed/memory evidence
> ---
>
> We agree practical scaling must be shown. We add end-to-end throughput/memory evidence:
>
> - **End-to-end ImageNet throughput.** On ImageNet-1K at $224^2$, swapping attention for LOTFormer improves wall-clock throughput on both scales: DeiT-Ti gains ~11% (2536.5 → 2816.4 img/s) and DeiT-S gains ~12% (940.4 → 1056.1 img/s), while also improving Top-1/Top-5, even in unfused PyTorch.
>
>     | Model | Params | Input | Throughput | Top-1 | Top-5 |
>     |---|---:|:---:|---:|---:|---:|
>     | DeiT-Ti | 5.7M | 224^2 | 2536.5 | 72.2 | 80.1 |
>     | DeiT-Ti + LOTFormer | 6.1M | 224^2 | 2816.4 | *74.8* | *81.3* |
>     | DeiT-S | 22M | 224^2 | 940.4 | 79.8 | 85.7 |
>     | DeiT-S + LOTFormer | 23.5M | 224^2 | *1056.1* | *80.1* | *85.9* |
>
> - **Peak memory scaling vs long sequences.** Peak memory from 4k to 128k vs FlashAttention-3 shows higher constant-factor overhead (~23–40%) in the unfused baseline, but **linear growth** with $n$, consistent with per-head $O(nr)$ for fixed $r\ll n$. The gap is thus an implementation-constant issue, motivating FA-style fusion/tiling in future work.
>
>     | Seq len | FA-3 (GB) | LOTFormer (PyTorch) (GB) | LOTFormer overhead vs FA-3 |
>     |---------|-----------|--------------------------|----------------------------|
>     | 4k      | 18.2      | 22.4                     | +23%                       |
>     | 8k      | 20.8      | 26.1                     | +26%                       |
>     | 16k     | 25.9      | 33.5                     | +29%                       |
>     | 32k     | 36.2      | 48.3                     | +33%                       |
>     | 64k     | 56.8      | 78.0                     | +37%                       |
>     | 128k    | 98.0      | 137.3                    | +40%                       |
>
> ---
> ## Robustness of $\varepsilon$, Sinkhorn iterations, and rank $r$
> ---
>
> Ablations indicate stable behavior rather than brittle tuning:
> - **Pivot size $r$:** accuracy improves smoothly with $r$ and is stable near the optimum (Table 4 peaks at $r=32$ with only a small drop at $r=64$).
> - **Iterations $T$ and entropy $\varepsilon$:** performance rises from $T=1$ to $T=5$–$10$ then saturates (Table 6), so a small, task-agnostic $T$ suffices; accuracy is stable over wide $\varepsilon$ ranges.
>
> ---
> ## Causal masking
> ---
>
> Strict DS cannot hold under a causal (lower-triangular) mask, so LOTFormer is **not used for decoder causal self-attention**. We keep standard softmax there and apply LOTFormer only to bidirectional or cross-attention, and will clarify this scope.

---

> ### Comment · Reviewer_dxXq · 2025-11-28
>
> Thanks for your rebuttal. But I can only maintain my score.

---

### Author Response · Authors · 2025-11-21
**Global Response to Common Concerns**

We thank the reviewers for their thoughtful feedback. The recurring questions focus on: (i) LOTFormer’s novelty vs prior doubly-stochastic (DS)/Sinkhorn/linear attention, (ii) isolating LOTFormer from auxiliary tricks, (iii) practical efficiency and long-context/causal use, and (iv) OT-notation clarity (incl. the pivot distribution). We address these below and outline corresponding revisions/ablations.

### 1) Conceptual novelty vs. prior DS / Sinkhorn / linear attention
We clarify that LOTFormer is qualitatively different from prior DS and linear attention.
- **Prior DS attentions are not linear-time.** Sinkformer and ESPFormer enforce DS on dense $N\times N$ interactions: Sinkformer stays $O(N^2)$; ESPFormer is quadratic in training and only reaches $O(N\log N)$ after hard sorting at inference. LOTFormer is, to our knowledge, the first DS attention that is end-to-end $O(N)$ in training and inference.
- **First linear DS attention via low-rank OT factorization.** Standard linear attentions are fast but typically only row-stochastic. LOTFormer parameterizes attention as a low-rank OT factorization through a pivot measure, yielding an implicit closed-form DS matrix without forming any dense $N\times N$ object, while keeping $O(N)$ cost.
We will sharpen this contrast in Sec. 2–3.

### 2) OT notation, meaning of $\Gamma$, and closed-form LOTFormer attention
We will add a brief OT primer and shape table. Queries/keys define empirical measures $\mu,\nu$, and $\sigma$ is a learnable pivot over $r\ll n$ points.
- **$\Gamma$ meaning.** $\Gamma$ denotes an entropy-regularized OT coupling: a nonnegative plan with prescribed marginals. We solve $\Gamma^{(1)}\in U(p^0,p^1)$ between $\mu$ and $\sigma$, and $\Gamma^{(2)}\in U(p^0,p^2)$ between $\sigma$ and $\nu$.
- **Closed form.** The glued plan is $\Gamma=(\Gamma^{(1)})^\top\,\mathrm{Diag}(\sigma)^{-1}\,\Gamma^{(2)}$. In the balanced case, $A=\Gamma$ is doubly stochastic and
$\mathrm{LOTAttn}(Q,K,V)=AV=(\Gamma^{(1)})^\top(\mathrm{Diag}(\sigma)^{-1}(\Gamma^{(2)}V))$, computed without forming $n\times n$ attention.

### 3) Are ImageNet gains due to “extra tricks”?
Table 7 isolates auxiliary modules. They help all methods, but LOTFormer relies on them least:
- PolaFormer: 61.9 → 74.6 with DWC (**+12.7**).
- FlattenAttention: 71.8 → 74.1 with DWC (**+2.3**).
- LOTFormer, with our standard [CLS] handling, reaches 73.2, and the LOTFormer-specific add-ons are modest: CLS-only Pola gives **+0.4** (73.6) and DWC gives **+1.2** (74.8). Our softmax aggregation for [CLS] is **not an ad-hoc boost**, but a requirement of DS attention: in the low-entropy/high-rank regime, DS couplings can approach near-permutations, making the [CLS] row overly sharp and preventing true global aggregation. We therefore decouple [CLS] from DS balancing by using standard softmax for [CLS], while keeping the image-token rows DS.

### 4) Role of convolution in LRA
DWC is useful but not the main source of gains. LOTFormer without DWC averages **60.6** (Table 1); adding DWC gives **62.9**, mainly via Image (45.7 → 54.1). For fairness, PolaFormer without DWC gives avg **57.5** (image 42.0, listops 38.8, retrieval 79.5, text 59.5, pathfinder 67.5), still below LOTFormer w/o DWC. With matched DWC, PolaFormer$_{\alpha=3}$ is **60.72** vs LOTFormer + DWC **62.9**.

### 5) Causal masking compatibility
We appreciate the reviewers raising this. DS self-attention requires balanced row/column normalization, which conflicts with lower-triangular causal masks; thus DS methods (incl. LOTFormer) are not directly applicable to decoder-only causal self-attention without altering DS constraints. They remain suited for encoder self-attention and cross-attention. Accordingly, in NMT (Sec. 4.4) we use LOTFormer in the encoder and cross-attention, while keeping decoder self-attention causal. We will state this limitation explicitly.

### 6) Practical runtime/memory and FlashAttention
We apologize for confusion: our wall-clock plot does **not** include FlashAttention, and FlattenAttention is a separate linear baseline (not FlashAttention). We will add FlashAttention to the runtime figure under the same setup and report peak memory vs sequence length for FlashAttention, softmax, linear baselines, and LOTFormer.  LOTFormer is presently standard PyTorch (no fused Triton/CUDA kernels). Its linear, block-separable OT computations are compatible with chunking/fusion ideas, and we will explore a hardware-optimized implementation for the camera-ready.

### 7) Scaling beyond Tiny for ImageNet
We agree Tiny-only is limited. We are running ImageNet-1K at small and base ViT scales under the same protocol and will report these results (accuracy + efficiency) shortly.

We hope these clarifications and additional experiments address the reviewers’ questions and better highlight LOTFormer’s conceptual and practical contributions. We will answer the remaining questions one by one shortly.

---

### Author Response · Authors · 2025-11-23
**Global Response part 2**

In this second part of our global response, we provide additional experiments and analyses requested by the reviewers to further substantiate LOTFormer’s practical benefits. Specifically, we (i) extend the ImageNet-1K evaluation to DeiT-Small to confirm the accuracy–throughput gains across model scales, (ii) demonstrate plug-and-play replacement of self-attention on finetuned BERT models across GLUE tasks to test transferability beyond vision, and (iii) report peak-memory comparisons against FlashAttention to clarify the current unfused PyTorch overhead. Together, these results directly address the concerns about generality, end-to-end efficiency, and practical scaling raised in the reviews.

---
### DeiT-Small + LOTFormer results on Imagenet1k

| Model | Params | Input | Throughput | Top-1 | Top-5 |
|---|---:|:---:|---:|---:|---:|
| DeiT-Ti | 5.7M | 224^2 | 2536.5 | 72.2 | 80.1 |
| DeiT-Ti + LOTFormer | 6.1M | 224^2 | 2816.4 | *74.8* | *81.3* |
| DeiT-S | 22M | 224^2 | 940.4 | 79.8 | 85.7 |
| DeiT-S + LOTFormer | 23.5M | 224^2 | *1056.1* | *80.1* | *85.9* |

On DeiT-S, LOTFormer adds +1.5M params (22M → 23.5M) while improving throughput by ~12% (940.4 → *1056.1* img/s) and giving a reliable accuracy lift (Top-1 79.8 → *80.1*, Top-5 85.7 → *85.9*). Overall, LOTFormer shifts the accuracy–efficiency frontier favorably rather than trading one for the other.

---
### Plug-and-play on finetuned GLUE benchmark

In this plug-and-play conversion on BERT-base, we only replace the encoder self-attention blocks while keeping embeddings, FFNs, layer norms, and the task heads unchanged.

| Method | CoLA | SST2 | MRPC | STS-B | QQP | MNLI | QNLI | RTE | (%) Recover |
|---|---:|---:|---:|---:|---:|---:|---:|---:|---:|
| BERT-FT | 58.8 | 93.2 | 90.2 | 88.8 | 91.0 | 84.7 | 91.3 | 68.2 | 100.0 |
| T2R | 43.6 | 87.7 | 83.0 | 78.6 | 86.7 | 78.9 | 84.6 | 54.1 | 88.9 |
| T2R-HH | 56.9 | 90.9 | 89.1 | 77.7 | 90.0 | 77.4 | 84.5 | 56.3 | 93.5 |
| Hedgehog | 59.2 | 92.6 | 90.1 | 87.4 | 91.0 | 82.6 | 89.6 | 69.3 | 99.3 |
| **LOTFormer** | 59.1 | 92.5 | 90.3 | 89.7 | 90.5 | 83.0 | 90.2 | 68.5 | **99.7** |

---
### Peak memory vs FlashAttention

Our current LOTFormer is pure PyTorch (no fused Triton/CUDA), so we **don’t claim kernel-level parity with FlashAttention yet**. The key point is **algorithmic scaling**: LOTFormer builds a low-rank doubly-stochastic OT plan via pivoting, so each head uses two rectangular kernels $(n\times r,\ r\times n)$ with cost $O(nr)$ for fixed $r\ll n$, **never materializing an $n\times n$ attention map**. The *present* implementation is memory-heavier because these steps run as separate ops, autograd stores more intermediates, and DS balancing needs extra workspace. This is a constant-factor issue, not a scaling one. FA-style IO tricks transfer directly: tile the OT kernels to stay SRAM-resident, fuse (matmul → exp/temperature → DS norm → value mix), and recompute in backward instead of storing intermediates. A fused kernel should therefore move LOTFormer closer to FA in peak memory while preserving linear behavior, and we will explore these in future work. We include the table below **at the reviewers’ request**, to report peak memory of the current unfused PyTorch baseline and clarify the gap attributable to implementation constants.


| Seq len | FA-3 (GB) | LOTFormer (PyTorch) (GB) | LOTFormer overhead vs FA-3 |
|---------|-----------|--------------------------|----------------------------|
| 4k      | 18.2      | 22.4                     | +23%                       |
| 8k      | 20.8      | 26.1                     | +26%                       |
| 16k     | 25.9      | 33.5                     | +29%                       |
| 32k     | 36.2      | 48.3                     | +33%                       |
| 64k     | 56.8      | 78.0                     | +37%                       |
| 128k    | 98.0      | 137.3                    | +40%                       |

Across both vision and language settings, these added experiments reinforce the same takeaway: LOTFormer is a drop-in attention replacement that consistently improves the accuracy–efficiency tradeoff without relying on task-specific tuning. On ImageNet-1K, it yields reliable Top-1 gains and higher throughput on DeiT-Tiny and DeiT-Small, confirming scalability across model sizes. On GLUE, simply swapping self-attention preserves finetuned performance almost perfectly, demonstrating strong plug-and-play generality. Finally, while our current unfused PyTorch implementation has higher constant-factor memory than FlashAttention, the results and analysis clarify that LOTFormer’s algorithmic scaling remains linear, and that FA-style kernel fusions are a clear path to closing the remaining practical gap in future work.

---

### Meta-Review · Area_Chair_Z7As · 2025-12-22

**Summary:**

While the theoretical aspects of the paper are solid, its empirical and experimental setup has several major shortcomings, especially regarding scalability, clarity, and practical evaluation.
1. Inadequate Long-Sequence Evaluation: The paper's evaluation of linear attention on long sequences is insufficient, as the performance degrades with increasing sequence length. The LRA benchmark used is not convincing due to limited testing with only two transformer layers.
2. Limited Model Size: The experiments are done on small models, and there's no evaluation on large-scale models, limiting the evidence for scalability in real-world high-throughput settings.
3. LRA Benchmark: The LRA benchmark is not a strong indicator of model quality anymore, as other models perform much better. The 60% accuracy of LOTFormer doesn’t convincingly demonstrate its effectiveness.
4. Scaling Issues: The experiments are conducted at a small scale (Tiny), which does not reflect the model's scalability for larger, real-world tasks like ImageNet.
5. Efficiency vs. FlashAttention: It is unclear about the efficiency of the LOTFormer compared to FlashAttention. The paper doesn’t explain whether LOTFormer uses optimized kernels or chunkwise computation, and it is not clear how it performs with longer sequences.
6. Incremental Contribution: The contribution is incremental compared to previous linear attention and Sinkhorn-based methods. Gains on ImageNet are dependent on extra tricks, making the core benefit hard to isolate.
7. Speed and Memory: There is limited evidence of significant improvements in speed and memory usage, and there is no evaluation on long-context causal language modeling.

**Reviewer Concerns:**

Two reviewers clearly clarify that the authors did not fully address their concerns in the discussion, and the overall evaluation is negative.

**Reviewer Scores:**

The average rating is 3.50. No reviewer suggested raising their scores.

---

### Decision · Program_Chairs · 2026-01-26

Reject